# An electrochemically responsive B−O dynamic bond to switch photoluminescence of boron-nitrogen-doped polyaromatics

Baige Yang[1], Yu-Mo Zhang [1]✉, Chunyu Wang[1], Chang Gu[1], Chenglong Li [1]✉, Hang Yin [2]✉, Yan Yan[3], Guojian Yang[1] & Sean Xiao-An Zhang [1]✉

Boron-doped polycyclic aromatic hydrocarbons exhibit excellent optical properties, and regulating their photophysical processes is a powerful strategy to understand the luminescence mechanism and develop new materials and applications. Herein, an electrochemically responsive B−O dynamic coordination bond is proposed, and used to regulate the photophysical processes of boron-nitrogen-doped polyaromatic hydrocarbons. The formation of the B−O coordination bond under a suitable voltage is confirmed by experiments and theoretical calculations, and B−O coordination bond can be broken back to the initial state under opposite voltage. The whole process is accompanied by reversible changes in photophysical properties. Further, electrofluorochromic devices are successfully prepared based on the above electrochemically responsive coordination bond. The success and harvest of this exploration are beneficial to understand the luminescence mechanism of boron-nitrogen-doped polyaromatic hydrocarbons, and provide ideas for design of dynamic covalent bonds and broaden material types and applications.

Boron-doped polycyclic aromatic hydrocarbons (B-PAHs) exhibit excellent optical properties, such as high color purity and high luminescent efficiency. Recently, they have been studied as the booming material for their potential applications in bioimaging[1], anionic sensing[2], catalysis[3], supramolecular self-assembly[4], photoelectric materials[5–8], and so on. Regulating the photophysical processes of B-PAH derivatives is a powerful strategy to understand the luminescence mechanism and develop new materials and applications[9–13]. For example, a plane-to-bowl conversion accompanied by a significant electronic perturbation has been reported when the coordination number of the boron was changed from tri-coordination to tetra-coordination induced by the addition of an external Lewis base[14–16]. The new complex can be detected by adding phosphorus-embedded π-system to the borole-embedded polycyclic π-system, which exhibited interesting behavior in response to photoirradiation, along with

dynamic B−P bond in excited state[17]. Dual-emission properties of compounds that consist of an electron-donator and an electron-accepting triarylborane could be controlled by regulating the dynamic B−X (X: O, S, N, etc.) bond in an excited state under different temperature[18–20].

Dynamic covalent bonds, which can be formed and broken reversibly in response to external stimuli[21–24], have gained increasing research interest due to their potential application in materials science[25–29], organic synthesis[30–32], and so on. They have been extensively studied, including mechanical forces-responsive bond[33–36], pH-responsive bond[37–40], as well as non-invasive light-responsive bond[41–45], heat-responsive bond[46–50]. Compared with the above stimulation methods, electricity-responsive bonds have the advantages of good controllability, and have been studied into catalytic reactions based on metal coordination complexes[51,52], artificial molecular machines[53–55],

¹State Key Lab of Supramolecular Structure and Materials, College of Chemistry, Jilin University, Changchun, P. R. China. ²Institute of Atomic and Molecular Physics, Jilin University, Changchun, P. R. China. ³College of Instrumentation & Electrical Engineering, Jilin University, Changchun, P. R. China. ✉e-mail: zhangyumo@jlu.edu.cn; chenglongli@jlu.edu.cn; yinhang@jlu.edu.cn; seanzhang@jlu.edu.cn

molecular switches[56], electrochromism[57] and electrofluorochromism[58,59], and other fields. However, there are few researches on electrical regulation of boron bonds. Method of in situ reversible electro-regulated photophysical process of B-PAHs is still to be developed, and limited to explore the optical properties and expand the application of those materials.

Herein, an electrochemically responsive reversible B−O dynamic coordination bond is proposed, and the photophysical properties of B,N-PAHs are successfully regulated by electrochemical stimulation. The related system is composed of DtBuCzB with simple structure and *p*-benzoquinone (*p*-BQ) with excellent electrochemical redox properties. The mechanism of electrochemically responsive B−O coordination and photophysical processes is confirmed by NMRs, mass spectra, electrochemical cyclic voltammetrys, transient and steady absorption spectra and emission spectra experiments, combined with theoretical calculations. Further, an electrofluorochromic (EFC) system with short switching time and excellent reversibility is developed by utilizing the electrochemically responsive coordination bond, and device prototypes are also successfully fabricated.

## Results and discussion

B,N-PAHs have both desirable properties such as high color purity, narrow emission spectrum, and potentially high luminous efficiency[60-64]. Their fascinating optical properties are mainly due to its excellent symmetry and rigid conjugated structure, and of course it is also related to the different electron-donating/withdrawing abilities of the boron and nitrogen involved in the conjugation. It is precisely the synergistic effects of the two types of heteroatoms on the lowest unoccupied molecular orbital (LUMO) and the highest occupied molecular orbital (HOMO) and molecular polarization of the conjugated molecule that determine its photophysical properties[65-69]. That means, how to adjust the electron distribution around the nitrogen and boron is the key to the change of the photophysical properties of B,N-PAHs under electrochemical stimulation. In this work, DtBuCzB (Fig. 1a) with narrow full-width at half-maximum (FWHM) and high color purity were selected firstly as an example[70]. The unoccupied orbital makes the boron atom in DtBuCzB act as a good electron acceptor in conjugation. Herein, *p*-benzoquinone (*p*-BQ) was chosen as an electrochemically regulated electron donor due to its remarkable change of electric charge density before

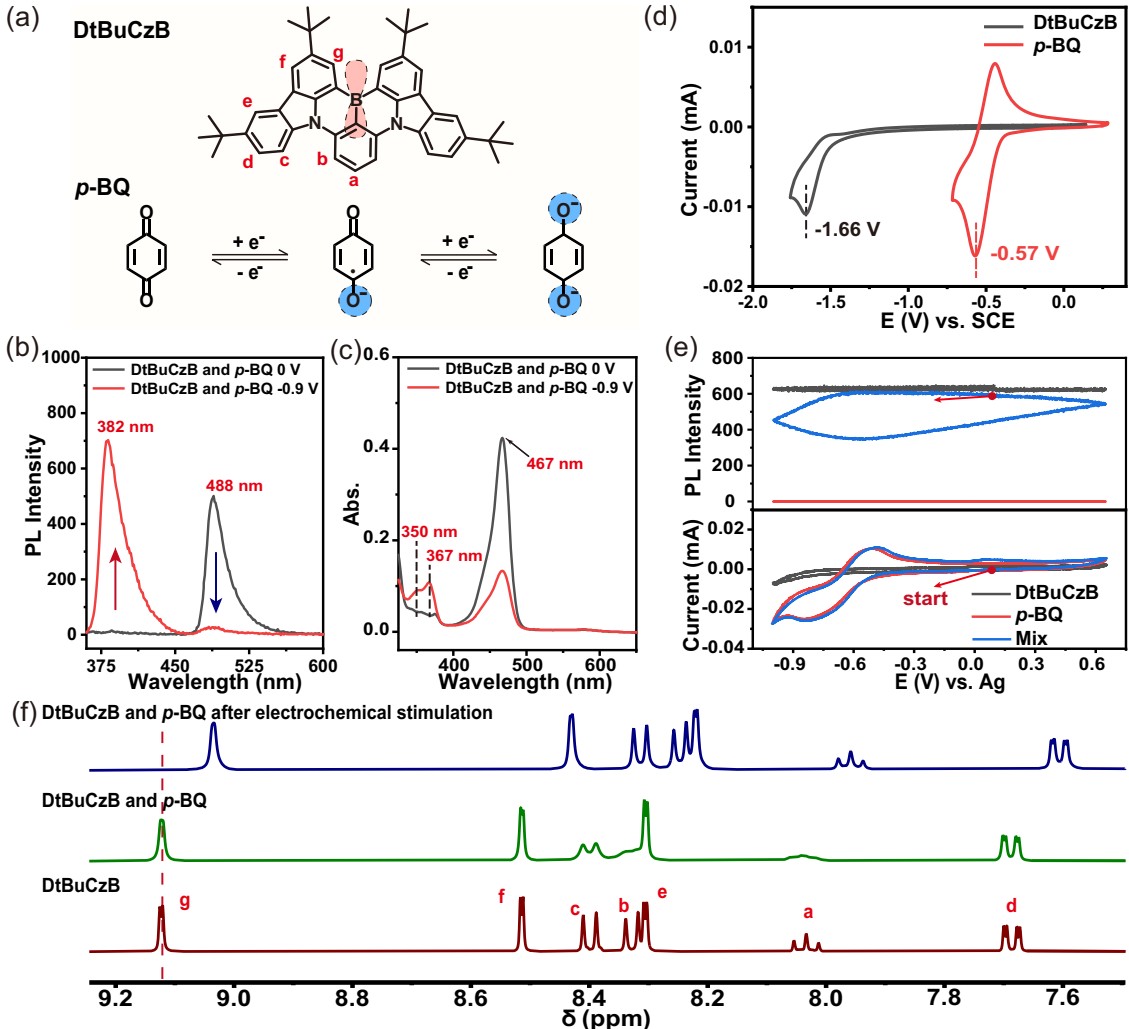

**Fig. 1 | Photophysical characteristics of the electrochemically responsive dynamic system of DtBuCzB and *p*-BQ. a** Molecular structures of DtBuCzB and *p*-BQ. The emission (**b**) and absorption spectra (**c**) of the mixture of *p*-BQ ($1.0 \times 10^{-3}$ mol L$^{-1}$) and DtBuCzB ($1.0 \times 10^{-4}$ mol L$^{-1}$) in THF with $1.0 \times 10^{-1}$ mol L$^{-1}$ tetrabutylammonium hexafluorophosphate (TBAPF$_6$) when the solutions were added 0 V and −0.9 V in situ, respectively. **d** Cyclic voltammograms of DtBuCzB ($1.0 \times 10^{-3}$ mol L$^{-1}$) and *p*-BQ ($1.0 \times 10^{-3}$ mol L$^{-1}$) in THF with $1.0 \times 10^{-1}$ mol L$^{-1}$ TBAPF$_6$.

**e** Changes in emission spectra at 488 nm (top) during cyclic voltammograms (CVs, bottom) in situ of DtBuCzB ($1.0 \times 10^{-4}$ mol L$^{-1}$), *p*-BQ ($1.0 \times 10^{-3}$ mol L$^{-1}$), and the mixture of DtBuCzB and *p*-BQ ($1.0 \times 10^{-4}$ mol L$^{-1}$ and $1.0 \times 10^{-3}$ mol L$^{-1}$) in THF with $1.0 \times 10^{-1}$ mol L$^{-1}$ TBAPF$_6$, ex = 440 nm. **f** $^1$H-NMR spectra of DtBuCzB ($1.25 \times 10^{-2}$ mol L$^{-1}$) and the mixtures of DtBuCzB and *p*-BQ ($1.25 \times 10^{-2}$ mol L$^{-1}$ and $1.5 \times 10^{-1}$ mol L$^{-1}$) in CD$_2$Cl$_2$ before and after electrochemical stimulation recorded at 400 MHz at room temperature, with tetramethylsilane as the internal standard.

**Table 1 | Association constant $K_a$ value[a] of DtBuCzB and *p*-BQ system before and after electrochemical stimulation at 298 K**

| | DtBuCzB and *p*-BQ | DtBuCzB and *p*-BQ under electricity |
|---|---|---|
| $K_a/M^{-1}$ | $210 \pm 7.1$ | $26{,}000 \pm 5700$ |

[a]The $K_a$ value is obtained from the emission spectra in THF.

and after electroreduction and excellent electrochemical redox performance[71–73].

First, the photophysical properties of a mixed system of DtBuCzB and *p*-BQ under electrochemical stimulation were investigated. As shown in Fig. 1b, c, the photophysical properties of the above system were obviously changed under negative voltage stimulation. To be specific, when a negative voltage (−0.9 V) was applied, the emission peak at 488 nm and absorption peak at 467 nm of DtBuCzB both decreased significantly, while new emission peak at 382 nm and new absorption peaks at 350 nm and 367 nm appeared, along with the disappearance of the fluorescence and color in the visible range. The actual pictures were shown in Supplementary Fig. 1. However, the absorbance and fluorescence of alone *p*-BQ or DtBuCzB did not exhibit similar change under the same stimulative voltage (−0.9 V, Supplementary Fig. 1). This phenomenon showed that the interaction between DtBuCzB and *p*-BQ was the key to realize the change of photophysical properties. A similar phenomenon was observed when 4-dimethylaminopyridine (DMAP) was added to DtBuCzB as shown in Supplementary Fig. 2, which has been proved from a coordination between them driven by the Lewis acid-base reaction. Hence, it is hypothesized that under electrochemical stimulation, there was an electrochemically regulated coordination between DtBuCzB and *p*-BQ, which led to significant changes in photophysical properties. The conjecture can be supported by adding the reference molecule sodium phenol to DtBuCzB, and the emission and absorption are significantly reduced (Supplementary Fig. 3).

In order to further explore the electrochemically switchable coordination between DtBuCzB and *p*-BQ, their electrochemical redox properties were investigated by cyclic voltammetry. As shown in Fig. 1d, the reduction potential of DtBuCzB was at −1.66 V, and *p*-BQ was at −0.57 V. The huge voltage gap between them ensured that *p*-BQ can be preferentially reduced at lower voltage stimulation when they are mixed. Furthermore, the changes of fluorescence at 488 nm and absorbance at 467 nm were studied by in situ tests, when DtBuCzB, *p*-BQ and their mixture solution were determined by cyclic voltammetry (Fig. 1e and Supplementary Fig. 4). Changes in emission and absorption can only occur when both *p*-BQ and DtBuCzB were present, and its reversibility depended on the redox of *p*-BQ. Compared with the irreversible electrochemical reduction of individual DtBuCzB (Supplementary Fig. 5), the hybrid system has the advantages of low redox potential and good reversibility. Furthermore, the association constants of DtBuCzB and *p*-BQ were calculated before and after electrochemical stimulation through emission spectra, and the obtained data are listed in Table 1 and Supplementary Fig. 6. The association constant ($26{,}000 \pm 5700$ M$^{-1}$) after electrochemical stimulation was two orders of magnitude higher than the initial one ($210 \pm 7.1$ M$^{-1}$), which further confirmed that the coordination between the above two substances occurred after electrochemical stimulation.

Then, electrochemically switchable coordination between two materials above was further characterized by $^1$H-NMR. As original *p*-BQ was added to DtBuCzB, the characteristic peak of DtBuCzB did not change significantly, indicating that there was no interaction between them (Fig. 1f). Interestingly, the characteristic peak of DtBuCzB moved towards the high field when the product generated by the electroreduction of *p*-BQ was added (Fig. 1f). It means that electroreduction product of *p*-BQ increased the electron cloud density around boron atom, demonstrating the formation of electrochemically switchable

B−O bond. The conclusion was supported when the phenol anion (BQ$^-$) from sodium phenol as reference was added to DtBuCzB (Supplementary Fig. 7). Further, as shown in Supplementary Figs. 7 and 8, the chemical shift also was observed when the phenol anion (BQ$^-$) from sodium phenol or normal Lewis base DMAP was added, resulting from coordination between BQ$^-$ or DMAP and DtBuCzB dominated by Lewis acid-base reaction. These accordant phenomena further verified that the formation of B−O coordination bond between *p*-BQ and DtBuCzB under electrochemical stimulation.

Hydroquinone di-lithium salt was used as a mimic molecule to further demonstrate the formation of B−O coordination bonds. Hydroquinone di-lithium salt was synthesized according to the method described in Supplementary Information, and its structure was proved by $^1$H-NMR and FT-IR spectra (Supplementary Figs. 17 and 20). As shown in Fig. 2a, b, a new set of characteristic signals appeared obviously when 0.5 eq. hydroquinone dianion was added to DtBuCzB, indicating that a portion of DtBuCzB coordinated with hydroquinone dianion, and new complexes, such as mono-coordination complex and di-coordination complex, had been produced. Then, with the gradual increase of the amount of hydroquinone dianion, the characteristic peaks belonging to the free DtBuCzB disappeared, and the characteristic peaks tended to exist in the mono-coordination complex. Those phenomena proved that the coordination between them occurred. In particular, when excess hydroquinone dianion was added, the signal of characteristic hydrogen (g) appeared the new splitting, which was due to the configuration of DtBuCzB changed (sp$^2$ to sp$^3$) caused by the coordination, resulting in the different chemical environment for $g_1'$ and $g_2'$.

Meanwhile, as shown in Fig. 2c, the free DtBuCzB and the mixture of DtBuCzB and excess di-anion were characterized by $^{11}$B-NMR, and the obvious difference of the signals proved that the boron atom changed from sp$^2$ to sp$^3$ hybridization[74]. At the same time, DtBuCzB were mixed with different proportions of hydroquinone dianion (from hydroquinone di-lithium salt) for mass spectrometry. As shown in Supplementary Fig. 9, when a small amount of hydroquinone dianion was added, [Li(DtBuCzB·*p*-BQ)]$^-$ and [(DtBuCzB)$_2$·*p*-BQ]$^{2-}$ can be found in the mixed system, in which the di-coordination was dominant. When excessive hydroquinone dianion was added, the presence of [Li(DtBuCzB·*p*-BQ)]$^-$, [H(DtBuCzB·*p*-BQ)]$^-$ and [(DtBuCzB)$_2$·*p*-BQ]$^{2-}$ can still be observed in the mixed system, which is dominated by mono-coordination. The above phenomenon was consistent with the $^1$H-NMR of DtBuCzB with different equivalents of hydroquinone dianion, which proved the formation of B−O coordination bonds in complex. Those results proved the formation of B-coordination bonds.

Meanwhile, the formation of electrochemically switchable B−O coordination bond was explored in depth using cyclic voltammetry. As shown in Fig. 3a, b, in the absence of DtBuCzB, the electrochemical behavior of *p*-BQ in tetrahydrofuran was classic two successive one-electron quasi-reversible process[75]. Among them, the peaks Ic and Ia respectively corresponded to the reduction of *p*-BQ to the semiquinone anion radical (*p*-BQ$^{\bullet-}$) and the oxidation of *p*-BQ$^{\bullet-}$ back to *p*-BQ, defined as Process ①. The peaks IIc and IIa respectively corresponded to the reduction of *p*-BQ$^{\bullet-}$ further to the di-anions (*p*-BQ$^{2-}$) and the oxidation of *p*-BQ$^{2-}$ back to the *p*-BQ$^{\bullet-}$ defined as Process ②. With the addition of electron acceptor (DtBuCzB), the interaction occurred between them, and various possible reaction processes are depicted in Fig. 3b, where E1 to E6 correspond to the voltage value of the reduction potential of each reaction process, respectively. In detail, with DtBuCzB added, the first reversible redox wave Ic−Ia gradually became irreversible, while the new irreversible half-wave IIIa emerged at a more positive potential, accomplished by the second reversible reduction wave IIc−IIa became a less negative, wide wave and gradually disappeared. The above phenomenon was consistent with the electrochemical behavior of *p*-BQ derivative in the presence of proton donor acid[76–78], which have proved that those phenomena was

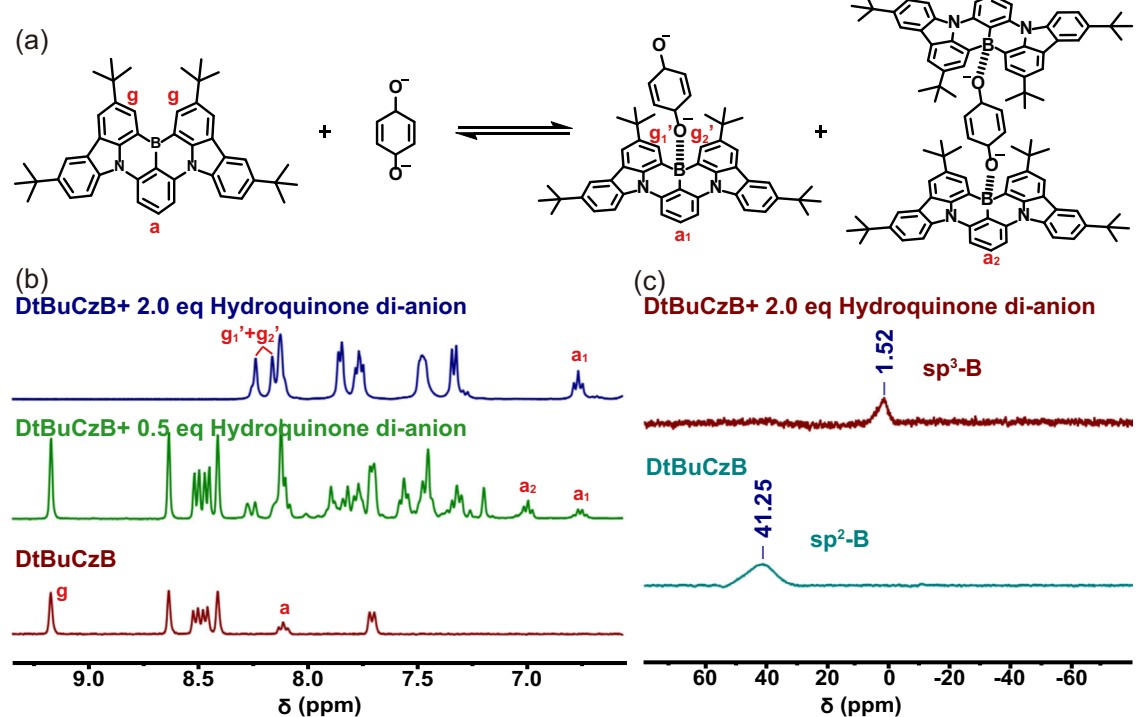

**Fig. 2 | Structural characterization of complexes. a** Reversible formation and dissociation of the B–O coordination bond. **b** $^{1}$H-NMR spectra of DtBuCzB, and the mixtures of DtBuCzB and different equivalent hydroquinone dianion in THF-$d_8$

recorded at 400 MHz at room temperature. **c** $^{11}$B-NMR (193 MHz) spectra of DtBuCzB and the mixture of DtBuCzB and hydroquinone dianion at 298 K in THF-$d_8$. BF$_3$·OEt$_2$ was used as an external standard (0 ppm).

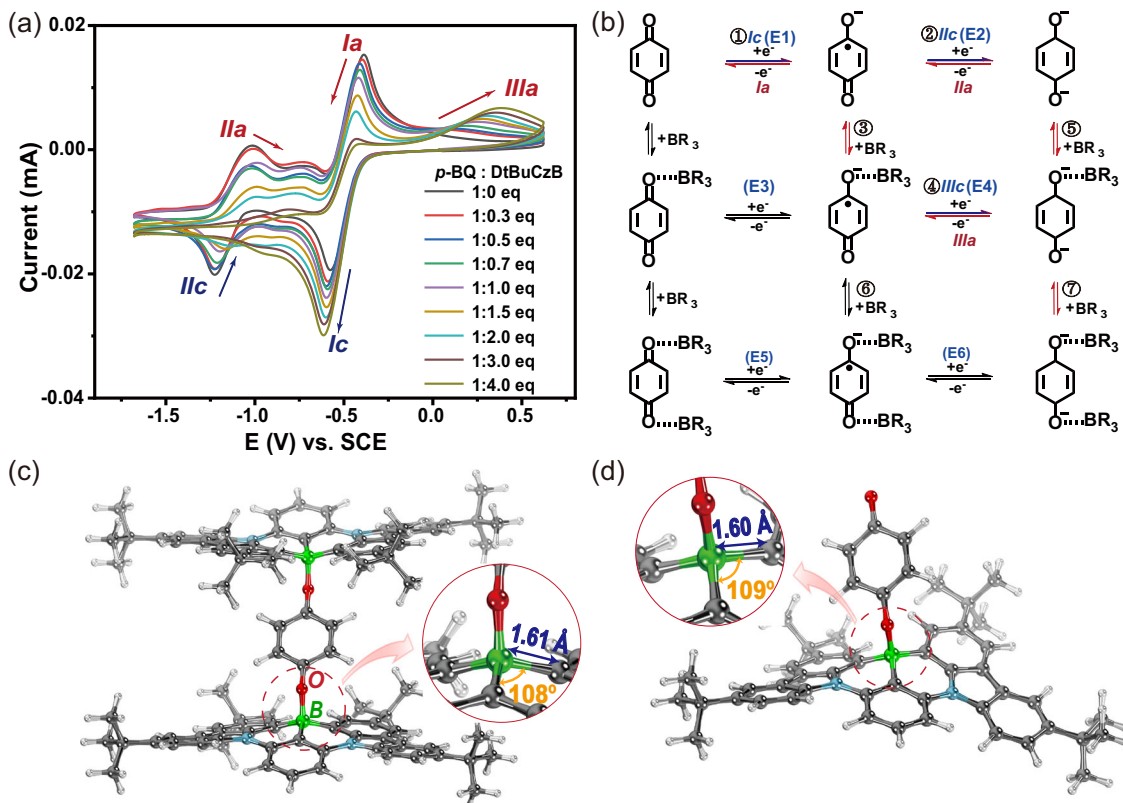

**Fig. 3 | Electrochemical mechanism and geometric structure of the electrochemically responsive dynamic system of DtBuCzB and $p$-BQ. a** Cyclic voltammograms of the mixture of $p$-BQ ($1.0 \times 10^{-3}$ mol L$^{-1}$) with different equivalents of DtBuCzB in THF with $1.0 \times 10^{-1}$ mol L$^{-1}$ TBAPF$_6$. All the scan rates are 50 mV/s.

**b** Schematic diagram of the mechanism of electrochemical reduction of the mixed system consisting of $p$-BQ and DtBuCzB, where BR$_3$ stands for DtBuCzB. Optimized molecular structure, bond length and bond angle of (**c**) [(DtBuCzB)$_2$·$p$-BQ]$^{2-}$ and **d** [DtBuCzB·$p$-BQ]$^{2-}$ complex by theoretical calculation.

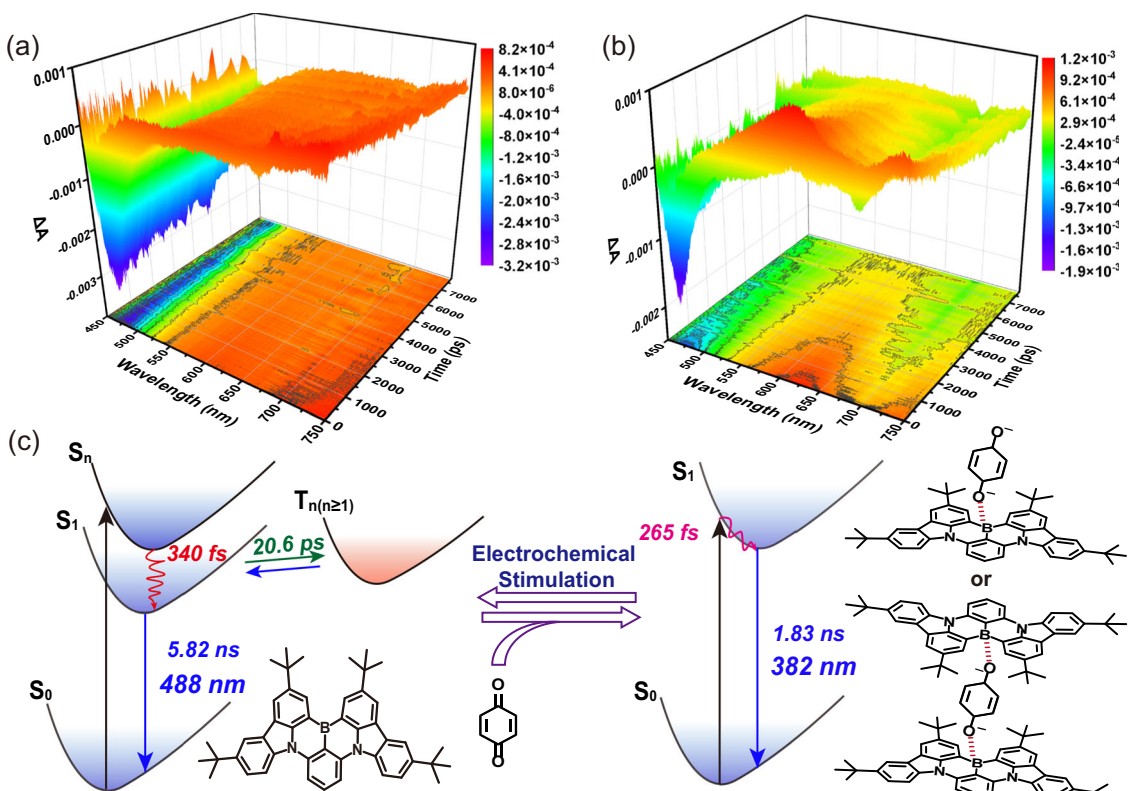

**Fig. 4 | Excited-state electron dynamics and mechanism of the electrochemically responsive dynamic system composed with DtBuCzB and *p*-BQ.** The transient absorption of the mixture of *p*-BQ ($1.0 \times 10^{-3}$ mol L$^{-1}$) and DtBuCzB ($1.0 \times 10^{-4}$ mol L$^{-1}$) with $1.0 \times 10^{-1}$ mol L$^{-1}$ TBAPF$_6$ (**a**) before and (**b**) after electrochemical stimulation in THF after excitation at 350 nm. **c** The mechanism of

photophysical property change of DtBuCzB based on electrochemically responsive B−O coordination bonds. The energy plots show the excited-state electron dynamics of the mixed system composed with DtBuCzB and *p*-BQ before and after electrochemical stimulation.

due to the complexation of *p*-BQ$^{\bullet-}$/*p*-BQ$^{2-}$ with H$^+$ through the Process ①, ③, ④, and ⑦ (Fig. 3b). Herein, the emergency of new irreversible half-wave IIIa illustrated that complexation occurred only after *p*-BQ has been reduced to the semiquinone radical (*p*-BQ$^{\bullet-}$) (Process ①, ③). Under these conditions, [DtBuCzB·*p*-BQ]$^{\bullet-}$ was more easily reduced than *p*-BQ$^{\bullet-}$, which is affected by the introduction of the electron-absorbing effect of DtBuCzB that caused E4 > E2. Therefore, with the addition of DtBuCzB, Process ④ and ⑦ gradually replaced Process ②, making the potential of reduction peak IIc gradually less negative. When a large amount of DtBuCzB was added, the IIc will have a fusion with the Ic, which may be the reason of the increase in the peak value of Ic. At the same time, differential pulse voltammograms of oxidation peak IIIa and reduction peak Ic, and their peak area proved this phenomenon. (Supplementary Fig. 10). This behavior was also observed for the reduction of 5H-benzo[b]carbazole −6,11-diones (BCDs) in the presence of phenol as a proton donor.[75] The above phenomenon supported that the [(DtBuCzB)$_2$·*p*-BQ]$^{2-}$ and [DtBuCzB·*p*-BQ]$^{2-}$ complex is more favorable than [DtBuCzB·*p*-BQ]$^{\bullet-}$ complex.

Then, geometric structures and absorption peaks of DtBuCzB, [DtBuCzB·*p*-BQ]$^{\bullet-}$, [(DtBuCzB)$_2$·*p*-BQ]$^{2-}$ and [DtBuCzB·*p*-BQ]$^{2-}$ complex were optimized by performing at o3lyp/TZVP level by the ORCA 5.0.3 package using the conductor-like polarizable continuum model (CPCM)[79,80]. As shown in Supplementary Tables 1, 4–9, after structural optimization of various possible structures, we speculated that [(DtBuCzB)$_2$·*p*-BQ]$^{2-}$ with theoretical absorption peaks at 372 nm and [DtBuCzB·*p*-BQ]$^{2-}$ with theoretical absorption peaks at 343 nm were determined to be plausible, corresponding to the absorption peaks at 367 nm and 350 nm in the experimental data. In the optimized geometric structure (Supplementary Data 1–3), the length of B−C bond

undergone a significant change from 1.53 Å for DtBuCzB to 1.61 Å for [(DtBuCzB)$_2$·*p*-BQ]$^{2-}$ or 1.60 Å for [DtBuCzB·*p*-BQ]$^{2-}$, and the angle of C−B−C also changed from 115° for DtBuCzB to 108° for [(DtBuCzB)$_2$·*p*-BQ]$^{2-}$ or 109° for [DtBuCzB·*p*-BQ]$^{2-}$. (Fig. 3c, d and Supplementary Fig. 11). This phenomenon means that boron atom of DtBuCzB changed from sp$^2$ to sp$^3$ hybridization when coordinated with *p*-BQ$^{2-}$, along with the change in conjugation structure and the blue shift of absorption wavelength.

In order to speculate the internal mechanism for changing the photophysical properties of DtBuCzB by electrochemically switched B−O bond, in situ electrochemical ultrafine transient absorption (TA) spectroscopy was performed using 350 nm excitation. As shown in Fig. 4a and Supplementary Fig. 12, TA spectra showed a negative peak with minima at 470–475 nm, attributed to the stimulated emission (SE, S$_1$–S$_0$ transition) of DtBuCzB. Interestingly, a new characteristic peak at 625 nm was obviously observed when negative voltage was in situ applied (Fig. 4b), accompanied by a weakening of the signal of DtBuCzB's stimulated emission. The new peak can be attributed to the excited-state absorption of complex after electrochemical stimulation. Furthermore, the time constant of decay process of TA spectra was extracted through singular value decomposition (SVD) and global fitting techniques. Three decay components (340 fs, 20.6 ps, and 5.82 ns for DtBuCzB) and two decay components (265 fs and 1.83 ns for complex after electrochemical stimulation) can be obtained. Based on the obvious difference in time scale, different time constants could be reasonably assigned to different relaxation processes of excited states to decode the photodynamics of the system. First, an ultrafast time constant (340 fs) of DtBuCzB could be assigned to the internal conversion process from S$_n$ to S$_1$[81], due to the fact that the energy of the

applied (350 nm) far exceeded the optimal absorption energy of the molecule (467 nm), which was able to excite the molecule to a higher excited state ($S_n$). Second, the picosecond decays (20.6 ps for DtBuCzB) could be attributed to intersystem crossing, which caused the electron to jump from an excited singlet to an excited triplet. As for complex after electrochemical stimulation, the applied energy was matched with the absorbed energy, so internal conversion process was not involved for the complex. The ultrafast time constant (265 fs) could be assigned to the vibrational relaxation process, which redistributed the electrons to the lowest $S_1$ state. The final nanosecond decays (5.82 ns and 1.83 ns) were attributed to radiative fluorescence. Due to the limitation of time window (~8 ns) of ultrafast transient absorption, it was difficult to obtain accurate fluorescence lifetime[81,82]. The above information is helpful to better understand the luminescence mechanism of B, N-PAHs, master the method of regulating their photophysical properties, and provide a reference for the design of B, N-PAHs.

Based on the above experimental and theoretical results, the photophysical property regulation process of DtBuCzB under electrochemical stimulation could be speculated as follows (Fig. 4c). There was no interaction between DtBuCzB and p-BQ in the initial state, and the optical properties of the system were mainly contributed by DtBuCzB. The molecule transitioned from the ground state to the excited state, went through internal conversion, intersystem crossing, and finally returned to the ground state by radiative transition, along with fluorescence emission at 488 nm. When the electrochemical stimulation was applied, the electrochemically responsive B−O coordination bond was formed, and the energy level of the complex was changed. The excited-state electron went through vibrational relaxation, and then returned to the ground state by radiative transition, along with the blue-shifted fluorescence emission at 382 nm. In summary, the electrochemically regulated B−O coordination bond leads to a change in the photophysical properties of B,N-PAHs, and the process is completely reversible.

Moreover, the basic properties of fluorescence, fluorescence quantum efficiency ($\Phi_{PL}$) and fluorescence lifetime ($\tau$) of the system in different states were characterized with sodium phenol ($NaOC_6H_5$) and hydroquinone di-lithium salt ($Li_2O_2C_6H_4$) as mimics of product generated by the electroreduction of p-BQ, and the results are listed in Supplementary Table 2. With the addition of $OC_6H_5^-$ or $O_2C_6H_4^{2-}$, the $\Phi_{PL}$ of DtBuCzB at 488 nm dropped sharply to near zero, accompanied by the appearance of a new fluorescence peak near 366 nm or 368 nm. The universality of electrochemically regulated B−O coordination bonds had been explored further using the well-known DABNA (Supplementary Fig. 13). The photophysical properties of the mixed system consisting of DABNA and p-BQ can be reversibly regulated under electrochemical stimulation.

Electrofluorochromic (EFC) materials and devices with controllable fluorescence properties show great application potential in advanced anti-counterfeiting, information storage, and display[83–91]. Based on the above electrochemically switchable reversible B−O coordination bond mechanism, EFC devices containing DtBuCzB and p-BQ were fabricated and exhibited excellent EFC property (Fig. 5a). As shown in Fig. 5b, the fluorescence intensity began to decrease when −1.3 V was applied. As shown in Fig. 5c, the initial state of the device was fluorescent "On". After applying a negative voltage (−2.2 V), the fluorescence intensity at 490 nm decreased significantly, and its fluorescence returned to the initial state quickly after the voltage was removed. The fluorescence quenching efficiency ($\eta$) was greater than 90%, and the maximum contrast value ($I_{on}/I_{off}$) was 10.2. The corresponding photos were shown in Fig. 5d. Furthermore, EFC device exhibited a surprising switching time. It only took 0.4 s to quench 90% of the initial state, and it only took 0.8 s to recover to 90% of the initial state (Fig. 5e). The basic properties of EFC device before and after electrochemical stimulation were listed in Supplementary Table 3. In

addition, EFC device had good stability (Fig. 5f). The quenching efficiency only decreased by 10% after 1000 cycles, which was attributed to the mechanism of electrochemically regulated reversible coordination between electrochemically regulated electron donor (p-BQ) and DtBuCzB. Finally, in order to directly exhibit the potential application of EFC devices, the pattern display was achieved by etching. As shown in Fig. 5a, the fluorescence pattern was clearly visible under voltage stimulation.

In summary, we demonstrated an electrochemically responsive B−O dynamic coordination bond, which can be formed and broken reversibly under electrochemical stimulation. The photophysical processes of B,N-PAHs were regulated by utilizing dynamic coordination bond. Through the combination of experimental results and theoretical calculations, we have a detailed understanding about the formation of electrochemically responsive B−O coordination bond and the regulation of photophysical processes. Meanwhile, an EFC device prototype was fabricated successfully using the electrochemically responsive coordination bond. Finally, we believe that the study of photophysical processes regulated by electricity is beneficial to understanding the luminescence mechanism of B,N-PAHs, and can provide ideas for designing new materials. The dynamic bond proposed in this work can open up an approach for the development of dynamic switching systems, which has important implications for both the basic research of dynamic bond and the design of functional materials.

## Methods

### Electrochemistry
Cyclic voltammograms were measured by the three-electrode cell, consisted of a glass-carbon working electrode (3 mm), a Pt wire counter electrode and an Ag wire reference electrode. An H-type electrolytic cell including two-electrode (two Pt nets) was used for electrochemical reduction of p-BQ. The mixture of p-BQ ($2.0 \times 10^{-1}$ mol $L^{-1}$) and TBAPF$_6$ ($1.0 \times 10^{-1}$ mol $L^{-1}$) in $CD_2Cl_2$ was used as the working solution. The mixture of ferrocene ($2.0 \times 10^{-1}$ mol $L^{-1}$) and TBAPF$_6$ ($1.0 \times 10^{-1}$ mol $L^{-1}$) in $CD_2Cl_2$ was used as the counter solution to balance the charge. The reaction was stirred under −1.2 V in the glove box. Details are available in the Supplementary information.

### Fabrication of the electrofluorochromic (EFC) devices
The EFC solution was a mixture of PMMA (24.7%, wt%), TBAPF$_6$ (1.1%, wt%), 1,4-dutyrolactone (73.8%, wt%), electro-Lewis base (BQ-OCH$_3$) (0.358%, wt%) and DtBuCzB (0.042%, wt%) in tetrahydrofuran (THF). The ion storage solution was a mixture of PTMA-co-BP (10 mg/mL) in THF. First, the EFC film layer was deposited by drop coating on the first ITO glass in the glove box. Next, the ion storage film was deposited by spin coating on the second ITO glass (80 µL, 500 r.p.m., 30 s). Then, the ion storage layer was obtained from UV-crosslinking in the glove box (254 nm for 10 min). Finally, a two-layer EFC device was fabricated by assembling the two ITO glasses together.

### Calculation of association constant (Ka)
The association constant ($K_a$) of DtBuCzB with p-BQ (the stoichiometry of DtBuCzB·p-BQ is 1:1) was expressed as Eq. (1):

$$\frac{Flu_{[DtBuCzB]_0} - Flu_{obs}}{Flu_{[DtBuCzB]_0}}$$

$$= \frac{\left([DtBuCzB]_0 + [p-BQ]_0 + \frac{1}{K_a}\right) - \sqrt{\left([DtBuCzB]_0 + [p-BQ]_0 + \frac{1}{K_a}\right)^2 - 4[DtBuCzB]_0[p-BQ]_0}}{2 \times [DtBuCzB]_0}$$

$$(1)$$

Where $[DtBuCzB]_0$ is the concentration of the fix initial concentration of DtBuCzB, $[p\text{-}BQ]_0$ is the concentration of the initial concentration of p-BQ, $Flu_{obs}$ is the fluorescence intensity when complexation have been dynamic equilibrium, $Flu_{[DtBuCzB]_0}$ $Flu_{[DtBuCzB]_0}$ is the

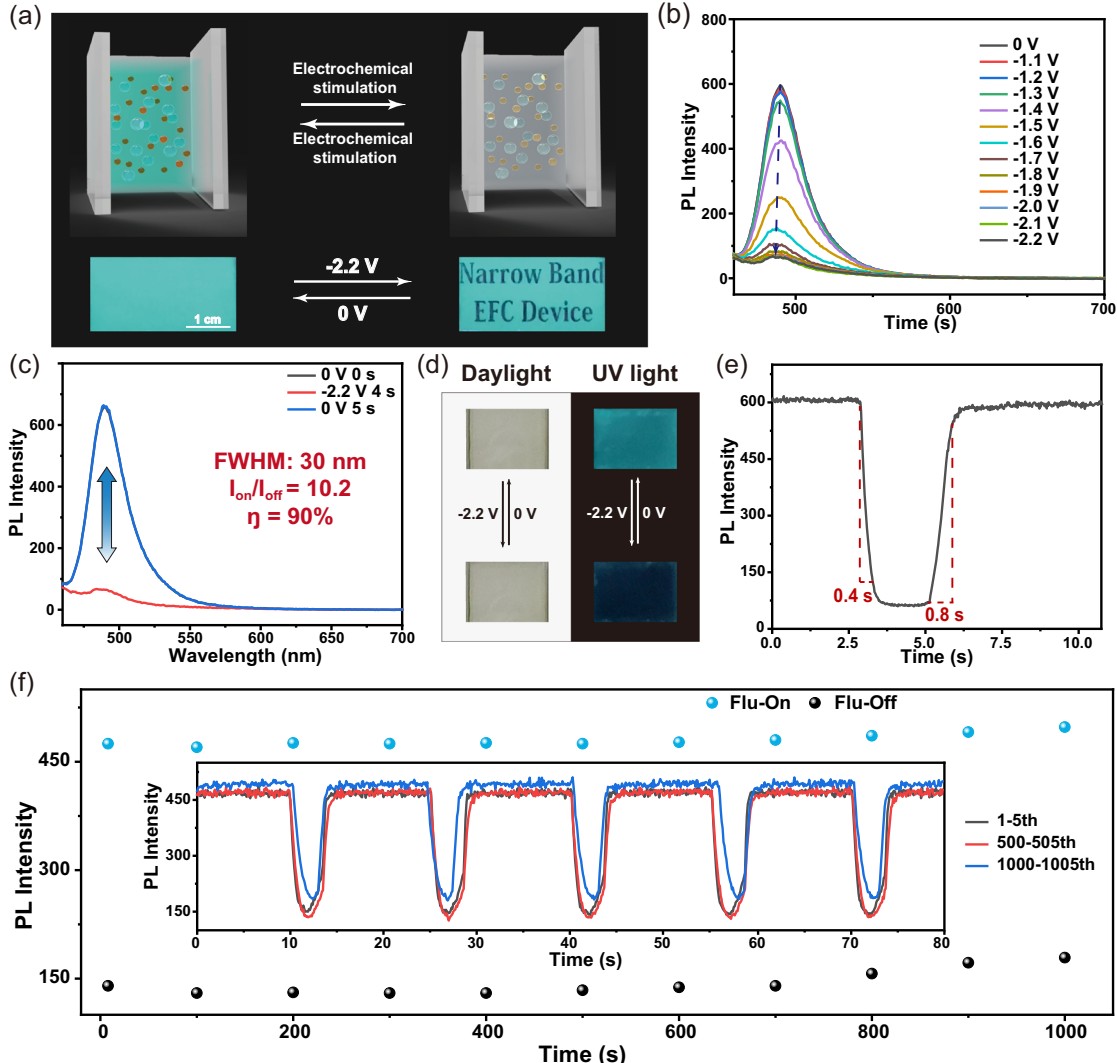

**Fig. 5 | Application of electrochemically responsive B–O dynamic bond on electrofluorochromic devices. a** Schematic diagram and actual pattern pictures of electrofluorochromic devices of DtBuCzB and *p*-BQ. **b** Emission spectra of the device under different voltages. **c** Reversible emission spectrum switching of the device and its corresponding EFC properties. **d** The actual photos of the device in daylight and UV light. **e** The switching time of the EFC device stimulated with −2.2 V and +1.6 V (ex = 440 nm). **f** The stability of the device at 490 nm under 1000 cycles, stimulated with −1.5 V 2.0 s for quenching, open-circuit voltage, and +1.2 V 1.5 s for recovering (ex = 440 nm).

---

initial fluorescence intensity of DtBuCzB, Computational details are available in the Supplementary information.

## Data availability
The data supporting the findings of this study are available within the article and the Supplementary Information and from the corresponding authors on request.

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

## Acknowledgements

This study was supported by the National Natural Science Foundation of China (Grant Nos. 22075098, Y.-M.Z.), the Jilin University Interdisciplinary Youth Innovation Team (Grant Nos. 45123031C002, Y.-M.Z.), and the Fundamental Research Funds for the Central Universities.

## Author contributions

Y.-M.Z., S.X.-A.Z., and B.Y. conceived this project, designed the experiments, and wrote and revised the manuscript. The manuscript was written through the contributions of all authors. B.Y. performed the

experiments. C. Wang provided Nuclear magnetic characterization. C.G., Y.Y., and G.Y. provided the counter electrode material (PTMA-co-BP). C.L. provided the DtBuCzB. H.Y. performed the theoretical calculations. All authors have given approval to the final version of the manuscript.

## Competing interests

The authors declare no competing interests.
