## [Peer Review File · Nature Communications]

An electrochemically responsive B–O dynamic bond to switch photoluminescence of boron-nitrogen-doped polyaromaticsREVIEWER COMMENTS

Reviewer #1 (Remarks to the Author):

Baige Yang and coworkers present a very interesting study on the reversible formation of B–O bonds by injection of electrons into para-benzoquinone. The formation and cleavage of the B–O bond results as expected in a drastic change in the photophysical properties. Albeit that the effect is expected, the strategy to induce this change via reversible reduction/oxidation of para-benzoquinone is novel. Furthermore, the principal applicability of this strategy for the fabrication of a device has been successfully demonstrated. The synthetic part and as far as this referee is able to judge the photophysical section have been both performed thoroughly. In summary, the conclusions are sound.

Reviewer #2 (Remarks to the Author):

The manuscript by Zhang and co-workers presents an electrofluorochromic system based on a boron, nitrogen-doped PAH. The authors showcase the dynamic covalent chemistry of the boron center, where B–O coordination bond formation leverages two distinct emissive species (i.e., with tri- and tetra-coordinate boron centers). The O-donor strength was proposed to be regulated by exploiting the redox equilibrium between benzoquinone, semiquinone, and hydroquinone. The excited-state dynamics were investigated through transient absorption spectroscopy. Additionally, electrofluorochromic devices with excellent switching time were demonstrated. While the results are interesting and would be appreciated by the field, the manuscript is difficult to understand and the writing/readability needs to be improved. With respect to novelty, the main boron-containing PAH (DtBuCzB) that is facilitating the binding is a known compound, thus I have mostly considered the electrofluorochromic system to be the main new contribution for consideration. With respect to that, some essential characterization data are missing, and the studies are incomplete (details below). I therefore conclude that the manuscript may become suitable publication, but after major revisions that include acquiring concrete experimental proof of the O-binding process. Below, I have detailed major concerns/comments as well as minor points that should be addressed.

Major comments:

1. My chief concern is the lack of characterization for this process and currently there is not conclusive experimental evidence for B-O bond formation. The authors have used DMAP as a probe to mimic O-binding to the boron center. However, in the opinion of this reviewer, this is not the best experiment to mimic the reaction of interest. DMAP features a relatively strongly binding N-donor and does not compare well with the O-donor. Furthermore, the authors state “Then, electric switchable coordination between two materials above was further characterized by ¹H-NMR. As original p-BQ was added to DtBuCzB, the characteristic peak of DtBuCzB did not change significantly, indicating that there was no interaction between them (Fig. 1f). Interestingly, the characteristic peak of DtBuCzB moved towards the

high field when the product generated by the electro-reduction of p-BQ was added (Fig. 1f). It's means that electro-reduction product of p-BQ increased the electron cloud density around B atom, demonstrating the formation of electric switchable B-O bond. And, the same phenomenon was observed when the phenol anion (BQ-) from sodium phenol as reference was added to DtBuCzB (Supplementary Fig. 5)." However, the same phenomenon is not observed when the phenol anion was added to DtBuCzB. In fact, Supplementary Fig 5 shows NMR spectra that are nearly identical. The very minor degree of shifting is negligible considering two compounds are now in solution (top spectrum, green). The resolution is also poor. I suggest considering the reaction of DtBuCzB with hydroquinone in the presence of an inorganic base for a more insightful analysis. The authors may also need to run variable temperature NMR to see if coordination of phenolate can actually be observed. I note that ^{11}B NMR spectroscopy is diagnostic for this type of interaction as the signals for tri- and tetra-coordinate boron centers are distinct. Thus, these boron NMR studies should be included and are standard in the boron PAH field.

2. The authors mentioned that "Compared with experimental and theoretical absorption data listed in Table S1, $[(\text{DtBuCzB})_2 \cdot \text{p-BQ}]^{2-}$ and $[\text{DtBuCzB} \cdot \text{p-BQ}]^{2-}$ complex were determined to be plausible." It is convincing that the involvement of a semiquinone radical species is likely to be ruled out from the computational results. However, it is unclear how the results align with the presence of $[(\text{DtBuCzB})_2 \cdot \text{p-BQ}]^{2-}$ and $[\text{DtBuCzB} \cdot \text{p-BQ}]^{2-}$.

3. Similarly, the authors presented the transient absorption spectra to explain the photophysical mechanism of "yet unidentified" electrochemically stimulated sample. What discussion is possible beyond "the energy level of complex has been changed" based on this experiment?

4. The authors should consider mass spectrometry to elucidate the formation of the suggested assemblies. DOSY NMR spectroscopy could also be an alternative.

5. The authors mentioned that "Compared with the above stimulation methods, there are few researches on dynamic covalent bonds in response to electrical stimulation, which have advantages of good controllability." There are a number of reports on redox-switchable coordination chemistry (J. Am. Chem. Soc. 2018, 140, 14590–14594). Additionally, I also wonder the authors have considered electrochemically switchable molecular machines.

6. I suggest "electric switchable B-O bond" to be revised as "electrochemically switchable B–O bond".

7. I am curious about the extent to which this strategy is applicable beyond this specific example (i.e., scope). If a similar approach is feasible for other boron-doped PAHs, including the well-known DABNA, it would attract broader interest.

Minor comments:

1. Rather than hyphen (-), please use en dash(–) to express minus sign or chemical bond.

2. In "p-BQ", "p" should be in italic.

3. In Fig. 2, please explain what E1–E6 means in detail.
4. In the caption for Table 1, the association constant “Ka” should be revised with “a” as subscript.
5. Page 7, Paragraph 1, Line 3: Table S1 -> Supplementary Table 1
6. Page 8, Paragraph 2, Line 8: radiation transition -> radiative transition
7. In References (including Supplementary References), the journal name for “Angew. Chem. In. Ed.” should be revised as “Angew. Chem. Int. Ed.”
8. In Supplementary Fig. 4, “Modle” should be revised to “Model”, and “NMR Titradition” should be revised to “Fluorescence Titration”. Please revise the caption for Supplementary Fig. 4b as well. It is confusing as (a) and (b) is essentially the same.
9. The caption for Supplementary Fig. 14 does not contain any information about temperature, phase (solution or solid), waveband (excitation energy), and etc.

Reviewer #3 (Remarks to the Author):

Reviewer #4 (Remarks to the Author):

The manuscript by Yang et al. reports on the modulation of the photophysical properties of a boron-nitrogen-doped polyaromatic hydrocarbons (DtBuCzB) through a combination of complexation chemistry and electrochemical processes. In particular, complexation of DtBuCzB with p-Benzoquinone (p-BQ) via the formation of a B-O bond, is modulated by the electrochemical reduction of p-BQ. The result is a system (it can be considered a dyad) that switches from one absorption/fluorescence state to another upon electrochemical reduction. Very interesting is the fluorescence switch from green (emission at 488 nm) to blue (emission at 382 nm) with more than 100 nm color modulation. This electrofluorochromic effect has been well supported by a mechanistic point of view, corroborated by both experiments and theory.

I believe that the manuscript is novel and it can pave the way for the design of new materials based on dynamic switching bond. It is worth to be published after some revisions according to the comments appended below.

The authors should show the effect of the electric stimulation on DtBuCzB at voltages < -1.66 V at which reduction of the DtBuCzB occurs.

The coordination effect of DMAP on the optical properties of DtBuCzB has been shown as a function of the concentration of DMAP. Is there any similar concentration-dependent effect with p-BQ?

In Supplementary Fig. 7 the effect of phenolate on the optical properties of DtBuCzB has been shown. However, the authors should show the entire emission spectral region as in Fig. 1b in order to understand whether the peak at 375 nm still arise upon quenching the one at 488 nm, as in the case of p-BQ.

The electrochemical mechanism should be better discussed. Specifically, the mechanistic explanation provided to justify the disappearance of the second reduction peak of pBQ, the concomitant increase of the intensity of the first reduction peak and the occurrence of peak IIIa, is not fully convincing. The authors stated that "It indicated that the complex of p- BQ \bullet^- and DtBuCzB was relatively easier to be further reduced than p-BQ alone, due to the electron-absorbing effect of the DtBuCzB." However, I believe that a change in the redox potential rather than a change of the intensity of the I_c redox peak should be observed. Indeed, the half-wave oxidation peak of the pBQ/DtBuCzB]2- complex is at higher potential than that corresponding to the pBQ alone. However, the corresponding half-wave reduction peak is not observed in the CV. Therefore, it seems that complexation occurs only after pBQ has been reduced to the monocation radical. Under these conditions, the second reduction process becomes progressively hampered due to complexation. When the complex is oxidized at higher potentials (half-wave peak IIIa), it is no longer stable and dissociates, releasing the free DtBuCzB which accumulates at the working electrode at each cycle, with the consequent increase of the intensity of the I_c peak. The peculiarity of this system is that its fluorescence switches between two different colors (from green to blue) upon the application of the electrical stimulation. It is not simply an electrofluorochromic quenching process. Therefore, the full characterization of the device should include the response of the system even at 382 nm.

A full characterization of the photophysical properties of free DtBuCzB and its complexes at different oxidation states should be provided (quantum yields in solution and in device, etc.)

Finally, photographs of the device in the different states should be provided.

Below are some minor comments:

1) In the introduction, please, take into account a very important work in the field of molecular structure changes induced by electrical stimuli, that is Kanazawa et al J. Phys. Chem. A 2014, 118, 6026–6033 doi.org/10.1021/jp5060588 which investigate changes in the photophysical properties induced by electrical stimuli in a fluoran-based molecule.

2) The first part of Results and discussion up to "...electrochemical redox performance" should be moved in the introduction.

3) Fig. 4 d. the pulse sequence is -1.5V for 2s and +1.5V for 1.5 but it seems that between one and the other there is a pause of over 10 s OFF. What is the potential in these 10 s? 0V or is it held at +1.2V?

4) The authors report quantitative data on the EFC device such as number of cycles, ON and OFF times, % contrast lost, but do not report the maximum contrast value. This could be added.

Reviewer #5 (Remarks to the Author):

I co-reviewed this manuscript with one of the reviewers who provided the listed reports as part of the Nature Communications initiative to facilitate training in peer review and appropriate recognition for co-reviewers.

A Point-by-point Response to the Reviewers' Comments

Here, the detailed point-by-point responses to the comments of reviewers' have been enclosed below.

To reviewer #1:

Comment:

Baige Yang and coworkers present a very interesting study on the reversible formation of B–O bonds by injection of electrons into para-benzoquinone. The formation and cleavage of the B–O bond results as expected in a drastic change in the photophysical properties. Alebit that the effect is expected, the strategy to induce this change via reversible reduction/oxidation of para-benzoquinone is novel. Furthermore, the principal applicability of this strategy for the fabrication of a device has been successfully demonstrated. The synthetic part and as far as this referee is able to judge the photophysical section have been both performed thoroughly. In summary, the conclusions are sound.

Response:

Dear reviewer, thank you very much for your time and effort to carefully review the manuscript and provide valuable guidance. We are deeply gratified and motivated to receive the enthusiastic help and high recognition of the top experts in this field carefully selected by the editor. We also look forward to more opportunities to receive your work guidance and possible cross-collaboration in future. At the same time, we will strive to further improve the quality of this work according to other reviews input to benefit more readers. Thanks again for your help.

To reviewer #2:

Comment:

The manuscript by Zhang and co-workers presents an electrofluorochromic system based on a boron, nitrogen-doped PAH. The authors showcase the dynamic covalent chemistry of the boron center, where B–O coordination bond formation leverages two distinct emissive species (i.e., with tri- and tetra-coordinate boron centers). The O-donor strength was proposed to be regulated by exploiting the redox equilibrium between benzoquinone, semiquinone, and hydroquinone. The excited-state dynamics were investigated through transient absorption spectroscopy. Additionally, electrofluorochromic devices with excellent switching time were demonstrated. While the results are interesting and would be appreciated by the field, the manuscript is difficult to understand and the writing/readability needs to be improved. With respect to novelty, the main boron-containing PAH (DtBuCzB) that is facilitating the binding is a known compound, thus I have mostly considered the electrofluorochromic system to be the main new contribution for consideration. With respect to that, some essential characterization data are missing, and the studies are incomplete (details below). I therefore conclude that the manuscript may become suitable publication, but after major revisions that include acquiring concrete experimental proof of the O-binding process. Below, I have detailed major concerns/comments as well as minor points that should be addressed.

Response:

Dear reviewer, thank you very much for taking your valuable time to review our manuscript very carefully. From your professional comments, we sincerely realize that we are fortunate to be communicating with a top expert who is familiar with this field. We cherish the opportunity very much to listen to your opinions and suggestions to help us to improve the quality of this work.

Thank you for your kind reminder. We apologize for the less perfect readability of the article due to the shortcomings in our English writing. We tried our best to improve the manuscript and made some changes to the manuscript. These changes will not influence the

content and framework of the paper. We sincerely appreciate your friendly suggestions to help us further improve the writing quality of this paper.

Thank you for accurately recognizing the innovation value of our work and for your very valuable feedback on improving the quality of our manuscript. According to your prompt, we have added B-NMR, variable temperature H-NMR, DOSY, and mass spectrometry to further explore the formation of B–O and their electrochemical mechanism as shown in the response of the following comments. And we have re-enriched the data of the electrofluorochromic (EFC) part to better demonstrate the characterization of this newly developed EFC material.

Accordingly, the revised **Fig. 5** now consist of emission spectra, contrast ratio, quenching efficient (η), switching time, stability, and photographs in the different states of EFC devices has been used to replace the original one.

Fig. 5| Application of electrochemically responsive B–O dynamic bond on electrofluorochromic (EFC) devices. (a) Schematic diagram and actual pattern pictures of electrofluorochromic devices of DtBuCzB and *p*-BQ. (b) Emission spectra of the device under different voltages. (c) Reversible emission spectrum switching of EFC device and its corresponding EFC properties. (d) The actual photos of the device in daylight and UV light. (e) The switching time of the EFC device stimulated with -2.2 V and $+1.6$ V (ex = 440 nm). (f) The stability of the device at 490 nm under 1000 cycles, stimulated with -1.5 V 2.0 s for quenching, open-circuit voltage, and $+1.2$ V 1.5 s for recovering (ex = 440 nm).

And, the relevant discussion ‘As shown in **Fig. 5b**, the fluorescence intensity began to decrease when -1.3 V was applied.’

and ‘the maximum contrast value is 10.2. The corresponding photos were shown in **Fig. 5d**.’

and ‘And, the basic properties of EFC device before and after electrical stimulation were

listed in **Supplementary Table 2**'

has been added on **Page 13** of **Manuscript**.

At the same time, the relevant basic performance of EFC device was enriched, such as: fluorescence quantum yields before and after electrical stimulation, fluorescence contrast, quenching efficiency (η), switching time, etc. The above information was summarized into **Supplementary Table 2** and added into **Page 35** in the revised **Supplementary Information**.

Supplementary Table 2. The summary of basic performance of EFC device.

	λ_{em} [nm]	$\Phi_{PL}^{a)}$	$\Phi_{PL}^{b)}$	Contrast	η	Switching time [s]
EFC Device	490	16.7%	1.0%	10.2	90%	t_{on} : 0.4
						t_{off} : 0.8

^{a)} Fluorescence quantum yield of initial device. ^{b)} Fluorescence quantum yield of device after electrical stimulation.

Meanwhile, the photophysical properties in solution has also been characterized, and been listed in the new **Supplementary Table 3** on **Page 36** in the revised **Supplementary Information**, such as: fluorescence quantum efficiency (Φ_{PL}) and fluorescence lifetime (τ) of EFC materials corresponding to B–O coordination or de-coordination.

Supplementary Table 3. The summary of photophysical properties of DtBuCzB (sp^2 -hybrid), DtBuCzB·OC₆H₅⁻ (sp^3 -hybrid), and DtBuCzB·O₂C₆H₄²⁻ (sp^3 -hybrid) in solution.

Solution	$\Phi_{PL, ex=440\text{ nm}}$	$\Phi_{PL, ex=340\text{ nm}}$	τ_p [ns]
DtBuCzB	91.49% (488 nm)	-	5.25
DtBuCzB·OC₆H₅⁻	0.35% (488 nm)	21.12% (368 nm)	3.85
DtBuCzB·O₂C₆H₄²⁻	0.18% (488 nm)	35.26% (366 nm)	3.70

And, the corresponding discussion

'Moreover, the basic properties of fluorescence, fluorescence quantum efficiency (Φ_{PL}) and fluorescence lifetime (τ) of the system in different states were characterized with sodium

phenol (NaOC_6H_5) and hydroquinone di-lithium salt ($\text{Li}_2\text{O}_2\text{C}_6\text{H}_4$) as mimics of product generated by the electro-reduction of *p*-BQ, and the results were listed in **Supplementary Table 3**. With the addition of OC_6H_5^- or $\text{O}_2\text{C}_6\text{H}_4^{2-}$, the Φ_{PL} of DtBuCzB at 488 nm dropped sharply to near zero, accompanied by the appearance of a new fluorescence peak near 366 nm or 368 nm.'

has been added on **Page 11** in the revised **Manuscript**.

Thank you again for your sincere help to our article and make our work better. And details are listed in the point-to-point response below.

Major comment 1:

*My chief concern is the lack of characterization for this process and currently there is not conclusive experimental evidence for B–O bond formation. The authors have used DMAP as a probe to mimic O-binding to the boron center. However, in the opinion of this reviewer, this is not the best experiment to mimic the reaction of interest. DMAP features a relatively strongly binding N-donor and does not compare well with the O-donor. Furthermore, the authors state “Then, electric switchable coordination between two materials above was further characterized by $^1\text{H-NMR}$. As original *p*-BQ was added to DtBuCzB, the characteristic peak of DtBuCzB did not change significantly, indicating that there was no interaction between them (Fig. 1f). Interestingly, the characteristic peak of DtBuCzB moved towards the high field when the product generated by the electro-reduction of *p*-BQ was added (Fig. 1f). It's means that electro-reduction product of *p*-BQ increased the electron cloud density around B atom, demonstrating the formation of electric switchable B-O bond. And, the same phenomenon was observed when the phenol anion (BQ $^-$) from sodium phenol as reference was added to DtBuCzB (Supplementary Fig. 5).” However, the same phenomenon is not observed when the phenol anion was added to DtBuCzB. In fact, Supplementary Fig 5 shows NMR spectra that are nearly identical. The very minor degree of shifting is negligible considering two compounds are now in solution (top spectrum, green). The resolution is also poor. I suggest considering the reaction of DtBuCzB with hydroquinone in the presence of an inorganic base for a more insightful analysis. The authors may also need*

to run variable temperature NMR to see if coordination of phenolate can actually be observed. I note that ^{11}B NMR spectroscopy is diagnostic for this type of interaction as the signals for tri- and tetra-coordinate boron centers are distinct. Thus, these boron NMR studies should be included and are standard in the boron PAH field.

Response:

Great thanks for your professional hint and reminder on our work. According to your very helpful suggestions, we have made corresponding experiments to further prove the formation of B–O bond in the system. The detailed experimental results are shown below.

First, thanks to your kind reminder, we realized that the low resolution of original **Supplementary Fig. 5** was due to the poor solubility of sodium phenol in CDCl_3 . Therefore, we screened the solvent, and finally selected deuterated tetrahydrofuran, which has a good comprehensive solubility of both DtBuCzB and sodium phenol. As shown in the revised **Supplementary Fig. 7**, the characteristic hydrogen signal of DtBuCzB changed obviously with the addition of sodium phenol. When a small amount of sodium phenol was added, a new set of characteristic signals in the high field region appeared obviously, indicating that a portion of DtBuCzB coordinated with sodium phenol. As the sodium phenol content increased to excess, the original characteristic signals disappeared, which means that all DtBuCzB were coordinated and existed in coordination form in the system. The above phenomenon proved that B–O coordination bond could be formed between DtBuCzB and sodium phenol, and the formation of B–O coordination bond between the DtBuCzB and the products after electrochemical reduction of *p*-BQ.

So, a revised **Supplementary Fig. 7** in **Page 16**

Supplementary Fig. 7. $^1\text{H-NMR}$ spectra of DtBuCzB, Sodium phenolate and the mixture of DtBuCzB and different equivalent Sodium phenolate in $\text{THF-}d_8$ recorded at 400 MHz at room temperature.

has replaced the original

Supplementary Fig. 5. $^1\text{H-NMR}$ spectra of DtBuCzB ($1.25 \times 10^{-2} \text{ mol L}^{-1}$) and the mixtures of DtBuCzB ($1.25 \times 10^{-2} \text{ mol L}^{-1}$) and sodium phenolate ($6.25 \times 10^{-2} \text{ mol L}^{-1}$) in CDCl_3 recorded at 400

MHz at room temperature.

Secondly, as your professional advices, the reaction of hydroquinone with DtBuCzB in the presence of inorganic bases was used to analyze the reaction of *p*-BQ and DtBuCzB under electrical stimulation. Considering the difficulty of solubility, hydroquinone di-lithium salt was prepared by the organic base *n*-butyllithium, as a simulation of hydroquinone bi-anion generated from the electrochemically reduction of *p*-BQ.

The detailed procedure was as follows: a stock solution of 2 eq *n*-butyllithium (2.5 mol/L in *n*-hexane) was added to a solution of hydroquinone (0.1 mol/L in THF) drop by drop under a nitrogen atmosphere at -78°C . After the reaction was completed, the solid was filtered, and the solid was repeatedly washed with anhydrous THF to remove the unreacted *n*-butyllithium, and the final product was obtained. $^1\text{H-NMR}$ and FT-IR spectra are shown in the **Supplementary Fig. 19 and Fig. 22**.

Supplementary Fig. 19. $^1\text{H-NMR}$ spectra of hydroquinone and the product (hydroquinone di-lithium) from hydroquinone and *n*-butyllithium in $\text{THF-}d_8$ recorded at 400 MHz at room temperature.

Supplementary Fig. 22. FT-IR spectra of hydroquinone (H₂Q) and hydroquinone di-lithium.

Compared with the initial hydroquinone, the characteristic peak of the product disappeared at 7.5 ppm, indicating that the proton hydrogen of hydroquinone was robbed by the strong organic base (n-butyllithium). In FT-IR spectra, hydroquinone showed the characteristic peak at 3261 cm⁻¹ and 1353 cm⁻¹, corresponding to the stretching vibration and bending vibration of -OH, respectively. Those peaks disappeared in hydroquinone di-lithium salt, which proved the deprotonation was complete.

After preparing successfully, the hydroquinone di-anion was used as a mimetic for the product of electrochemical reduction of *p*-BQ to mix with DtBuCzB. And they were characterized by ¹H-NMR, variable temperature NMR, and ¹¹B-NMR, respectively.

Fig. 2| Structural characterization of complexes. (a) Reversible formation and dissociation of the B–O coordination bond. (b) ¹H-NMR spectra of DtBuCzB, and the mixtures of DtBuCzB and different equivalent hydroquinone di-anion in THF-*d*₈ recorded at 400 MHz at room temperature. (c) ¹¹B-NMR (193 MHz) spectra of DtBuCzB and the mixture of DtBuCzB and hydroquinone di-anion at 25 °C in THF-*d*₈. BF₃·OEt₂ was used as an external standard (0 ppm).

As shown in **2b**, a new set of characteristic signals appeared obviously when 0.5 eq. hydroquinone di-anion was added to DtBuCzB, indicating that a portion of DtBuCzB coordinated with hydroquinone di-anion, and new complexes, such as mono-coordination complex and di-coordination complex, had been produced. Then, with the gradual increase of the amount of hydroquinone di-anion, the characteristic peaks belonging to the free DtBuCzB disappeared, and the characteristic peaks tended to exist in the single form. Those phenomena proved that the coordination between them occurred. The structure of DtBuCzB before and after coordination was further confirmed by COSY ¹H-NMR (**Supplementary Fig. 9**). In particular, when excess hydroquinone di-anion were added, the signal of characteristic hydrogen (g) appeared the new splitting, which was due to the configuration of DtBuCzB changed (sp² to sp³) caused by the coordination, resulting in the different chemical environment for g₁' and g₂'.

Supplementary Fig. 9. Correlation spectroscopy (COSY, 600 MHz) of DtBuCzB before and after coordination in THF- d_8 recorded at room temperature.

^{11}B -NMR spectroscopy has a diagnostic effect on this interaction, since the signal of the three- and four-coordinate boron centers is evident. As shown in **Fig. 2c**, the free DtBuCzB and the mixture of DtBuCzB and excess bi-anions were characterized by ^{11}B -NMR, and the obvious difference of the signals proved that B changed from sp^2 to sp^3 hybridization (*Chemical Reviews*, 1992, 92, 325-362). At the same time, variable temperature ^1H -NMR (**Supplementary Fig. 10**) of a mixture of DtBuCzB and 2 eq. hydroquinone di-anion has been explored, and a new characteristic hydrogen signal appeared at low temperature. Those results proved the formation of B-coordination bonds.

Supplementary Fig. 10. Variable temperature $^1\text{H-NMR}$ (500 MHz, $\text{THF-}d_8$) of the mixture consist of DtBuCzB and 2 eq. hydroquinone di-anion measured from 223 K to 293 K.

Based on the above experimental results, the new **Fig. 2**, **Supplementary Fig. 9** and **10** have been added to the revised manuscript. And the corresponding description

‘Hydroquinone di-lithium salt was used as a mimic molecule to further demonstrate the formation of B–O coordination bonds. Hydroquinone di-lithium salt was synthesized according to the method described in **Supplementary Information**, and its structure was proved by $^1\text{H-NMR}$ and FT-IR spectra (**Supplementary Fig. 19** and **Fig. 22**). As shown in **Fig. 2b**, a new set of characteristic signals appeared obviously when 0.5 eq. hydroquinone di-anion was added to DtBuCzB, indicating that a portion of DtBuCzB coordinated with hydroquinone di-anion, and new complexes, such as mono-coordination complex and di-coordination complex, had been produced. Then, with the gradual increase of the amount of hydroquinone di-anion, the characteristic peaks belonging to the free DtBuCzB disappeared, and the characteristic peaks tended to exist in the single form. Those phenomena proved that the coordination between them occurred. The structure of DtBuCzB before and after coordination was further confirmed by COSY $^1\text{H-NMR}$ (**Supplementary Fig. 9**). In particular, when excess hydroquinone di-anion were added, the signal of characteristic hydrogen (g) appeared the new splitting, which was due to the configuration of DtBuCzB changed (sp^2 to sp^3) caused by the coordination, resulting in the different chemical environment for g_1' and g_2' .

Meanwhile, as shown in **Fig. 2c**, the free DtBuCzB and the mixture of DtBuCzB and excess bi-anions were characterized by ^{11}B -NMR, and the obvious difference of the signals proved that B changed from sp^2 to sp^3 hybridization. At the same time, variable temperature ^1H -NMR (**Supplementary Fig. 10**) of a mixture of DtBuCzB and 2 eq. hydroquinone di-anion has been explored, and a new characteristic hydrogen signal appeared at low temperature. Those results proved the formation of B-coordination bonds.'

has been added on **Page 7** in **Manuscript**.

Major comment 2:

The authors mentioned that “Compared with experimental and theoretical absorption data listed in Table S1, $[(\text{DtBuCzB})_2\cdot p\text{-BQ}]^{2-}$ and $[\text{DtBuCzB}\cdot p\text{-BQ}]^{2-}$ complex were determined to be plausible.” It is convincing that the involvement of a semiquinone radical species is likely to be ruled out from the computational results. However, it is unclear how the results align with the presence of $[(\text{DtBuCzB})_2\cdot p\text{-BQ}]^{2-}$ and $[\text{DtBuCzB}\cdot p\text{-BQ}]^{2-}$.

Response:

We sincerely appreciate your valuable suggestion, which will made our article clearer and easier to understand. In this work, theoretical calculations are used in combination with experimental data to predict the structure of the material generated after electrical stimulation.

As shown in **Supplementary Table 1**, the method adopted for theoretical calculation has been verified by individual DtBuCzB, and its error was within 10 nm, which proved that the functional (o3lyp) is effective. Therefore, the above functional was used to calculate the possible structure of the system after electrical stimulation. After structural optimization of various possible structures, according to the theoretical absorption peak distribution listed in **Supplementary Table 1**, we speculate that $[(\text{DtBuCzB})_2\cdot p\text{-BQ}]^{2-}$ with theoretical absorption peaks at 372 nm and $[\text{DtBuCzB}\cdot p\text{-BQ}]^{2-}$ with theoretical absorption peaks at 343 nm were most likely to exist, corresponding to the absorption peaks at 367 nm and 350 nm in the experimental data. Furthermore, the results of ^1H -NMR (**Major comment 1**) and Mass

spectrometry (*Major comment 4*) were all in agreement with the theoretical calculation, which proved the reliability of it.

Here, thanks a lot for your very kind reminder, we realized that the way we presented the data in the original manuscript was less clear enough and would make it difficult for the reader to quickly understand. Therefore, in the revised manuscript, the results of the theoretical calculations in **Supplementary Table 1** were reordered according to energy levels in order to get a more intuitive conclusion.

So, a modified **Supplementary Table 1**

Supplementary Table 1. Experimental and theoretical photophysical properties of compounds that may be present in the system.

Compound	$\lambda_{\text{abs}}^{\text{a)}}$ /nm (f)	$\lambda_{\text{abs}}^{\text{b)}}$ /nm	$\lambda_{\text{ex}}^{\text{a)}}$ /nm (f)	$\lambda_{\text{ex}}^{\text{b)}}$ /nm
DtBuCzB	461 (0.55572)	467	491 (0.59515)	488
Complex^{c)}	-	350, 367		
	372 (0.15967)			
[(DtBuCzB)₂·p-BQ]²⁻	376 (0.05845)			
	391 (0.06464)			
	343 (0.11629)			
[DtBuCzB·p-BQ]²⁻	362 (0.03073)			
	370 (0.09653)			
	647 (0.01184)			
[(DtBuCzB)₂·p-BQ]⁻	926 (0.00310)			
	2013 (0.00524)			
	417 (0.09864)			
[DtBuCzB·p-BQ]⁻	423 (0.06608)			
	460 (0.02114)			

^{a)}Theoretical photophysical properties of compounds; ^{b)}Experimental photophysical properties of compounds; ^{c)}the complex was produced by in-situ electric stimulation of a mixture of DtBuCzB and p-BQ.

has replaced the original

Supplementary Table 1. Experimental and theoretical photophysical properties of compounds that may be present in the system.

Compound	$\lambda_{\text{abs}}^{\text{a)}}$ /nm	$\lambda_{\text{em}}^{\text{a)}}$ /nm	$\lambda_{\text{abs}}^{\text{b)}}$ /nm	$\lambda_{\text{em}}^{\text{b)}}$ /nm
----------	--	---------------------------------------	--	---------------------------------------

DtBuCzB	461	491	467	488
[(DtBuCzB)₂·p-BQ]²⁻	372, 391, 376			
[(DtBuCzB)₂·p-BQ]⁻	647, 2013, 926	-		
[DtBuCzB·p-BQ]²⁻	343, 370, 362	-		
[DtBuCzB·p-BQ]⁻	417, 423, 460	-		
Complex^{e)}	-	-	350, 367	382

^{a)}Theoretical photophysical properties of compounds; ^{b)}Experimental photophysical properties of compounds; ^{c)}the complex was produced by in-situ electric stimulation of a mixture of DtBuCzB and p-BQ.

And, in order to describe more clearly, a more detailed explanation on **Page 9** of **Manuscript**

‘As shown in **Supplementary Table 1** and **Table 4-9**, after structural optimization of various possible structures, we speculate that [(DtBuCzB)₂·p-BQ]²⁻ with theoretical absorption peaks at 372 nm and [DtBuCzB·p-BQ]²⁻ with theoretical absorption peaks at 343 nm were determined to be plausible, corresponding to the absorption peaks at 367 nm and 350 nm in the experimental data.’

has replaced the original

‘Compared with experimental and theoretical absorption data listed in **Table S1**, [(DtBuCzB)₂·p-BQ]²⁻ and [DtBuCzB·p-BQ]²⁻ complexes were determined to be plausible.’

Meanwhile, the detailed results of the theoretical calculations were listed in **Supplementary Table 4-9**, which is added on **Page 37-42** of **Supplementary Information**.

Supplementary Table 4. The absorption spectrum via transition electric dipole moments of DtBuCzB.

State	Energy [cm⁻¹]	Wavelength [nm]	fosc	T2 [au**2]	TX [au]	TY [au]	TZ [au]
1	21704	460.7	0.55572	8.42932	-2.90127	-0.00004	-0.10945
2	26853.4	372.4	0.00381	0.04671	-0.00001	0.21613	0.00001
3	27007.5	370.3	0.007307	0.08907	0.2213	0	-0.20023

Supplementary Table 5. The emission spectrum via transition electric dipole moments of DtBuCzB.

State	Energy [cm ⁻¹]	Wavelength [nm]	fosc	T2 [au**2]	TX [au]	TY [au]	TZ [au]
1	20383.6	490.6	0.595146	9.61213	3.09768	0.00004	0.12856

Supplementary Table 6. The absorption spectrum via transition electric dipole moments of [(DtBuCzB)₂·*p*-BQ]²⁻.

State	Energy [cm ⁻¹]	Wavelength [nm]	fosc	T2 [au**2]	TX [au]	TY [au]	TZ [au]
1	25581.1	390.9	0.064642	0.8319	0.62964	0.64801	0.12466
2	25884.7	386.3	0.002082	0.02648	-0.14329	0.0768	0.00692
3	26596.2	376	0.058448	0.72348	0.59402	-0.59112	-0.14555
4	26615.5	375.7	0.043737	0.54099	0.4977	-0.54155	-0.00182
5	26857.5	372.3	0.035911	0.44019	-0.4475	0.48	0.0976
6	26874.9	372.1	0.159668	1.9559	1.00096	-0.97398	-0.07312

Supplementary Table 7. The absorption spectrum via transition electric dipole moments of [DtBuCzB·*p*-BQ]²⁻.

State	Energy [cm ⁻¹]	Wavelength [nm]	fosc	T2 [au**2]	TX [au]	TY [au]	TZ [au]
1	16364	611.1	0.02084	0.41925	0.28161	-0.1796	0.5547
2	16851.4	593.4	0.016387	0.32015	-0.089	0.51961	-0.20551
3	20277.8	493.1	0.003141	0.05099	0.14311	0.06724	0.16122
4	21876.2	457.1	0.006433	0.09681	-0.20058	0.02613	-0.23642
5	24257.6	412.2	0.001245	0.0169	0.09279	-0.06058	0.06798
6	25504	392.1	0.002673	0.0345	-0.0435	-0.0205	-0.17942
7	25589.5	390.8	0.001351	0.01738	0.01436	0.1265	0.03423
8	26158.8	382.3	0.000267	0.00336	-0.04184	0.01741	0.03612
9	26998.6	370.4	0.096533	1.1771	0.07699	1.07643	0.11161
10	27610.2	362.2	0.030731	0.36642	0.58146	0.15331	-0.06942
11	27935.4	358	0.004372	0.05152	0.10589	-0.15706	0.12508
12	28860.2	346.5	0.016548	0.18877	-0.35303	-0.15801	0.19792
13	29157.7	343	0.116286	1.31296	-0.02519	1.13161	0.17827

Supplementary Table 8. The absorption spectrum via transition electric dipole moments of [(DtBuCzB)₂·*p*-BQ]⁻.

State	Energy [cm ⁻¹]	Wavelength [nm]	fosc	T2 [au**2]	TX [au]	TY [au]	TZ [au]
1	4967.9	2012.9	0.005239	0.34715	0.20878	0.54153	0.10152
2	5006.2	1997.5	0.00025	0.01641	0.08145	0.08802	0.0451
3	7082.9	1411.9	0.000042	0.00195	-0.039	0.01984	-0.00613
4	7210.3	1386.9	0.002263	0.10333	-0.04298	0.31613	0.03931
5	10774	928.2	0.001328	0.04057	0.07866	0.17331	0.06595
6	10804.1	925.6	0.003105	0.09463	0.18999	0.20283	0.13186
7	10969.6	911.6	0.00011	0.00331	0.0393	0.00543	0.04165
8	11047.5	905.2	0.002144	0.0639	-0.21477	0.12897	0.03379
9	15275	654.7	0.002722	0.05867	-0.1265	-0.18363	-0.09463
10	15450.3	647.2	0.011839	0.25226	-0.10325	-0.47434	-0.12883

Supplementary Table 9. The absorption spectrum via transition electric dipole moments of [DtBuCzB·p-BQ]⁻.

State	Energy [cm ⁻¹]	Wavelength [nm]	fosc	T2 [au**2]	TX [au]	TY [au]	TZ [au]
1	10555.5	947.4	0.001846	0.05759	-0.07638	0.16213	-0.15958
2	12366.8	808.6	0.000324	0.00864	-0.01963	0.06523	-0.06322
3	13849	722.1	0.000007	0.00017	0.00843	0.00724	0.0071
4	16366.8	611	0.000665	0.01337	-0.08618	-0.0361	-0.06814
5	16410.5	609.4	0.000232	0.00466	-0.01683	0.04001	0.05264
6	19813	504.7	0.004356	0.07238	-0.13611	0.14016	-0.18496
7	21783.7	459.1	0.021139	0.31947	0.20382	-0.34585	0.39789
8	22879.8	437.1	0.001324	0.01906	-0.03746	0.00836	-0.1326
9	23629.3	423.2	0.066075	0.92059	0.46563	-0.25663	0.7987
10	23993	416.8	0.098637	1.35341	0.56525	-0.44401	0.91475

Major comment 3:

Similarly, the authors presented the transient absorption spectra to explain the photophysical mechanism of “yet unidentified” electrochemically stimulated sample. What discussion is possible beyond “the energy level of complex has been changed” based on this experiment?

Response:

Thank you very much for providing this very helpful and professional advice on improving our work. Herein, transient absorption spectroscopy had several applications here as a powerful tool for monitoring kinetic reactions of excited states. First, the in situ electrochemical-transient spectroscopy of mix system demonstrated the formation of a new

complex after electrical stimulation, which was concluded by the appearance of a new characteristic peak at 625 nm after electrical stimulation, which can be attributed to the excited state absorption of the complex. As mentioned in the manuscript '*Interestingly, a new characteristic peak at 625 nm was obviously observed when negative voltage was in situ applied (Fig. 4b), accompanied by weakening of signal of DtBuCzB's stimulated emission. The new peak can be attributed to the excited state absorption of complex after electrical stimulation.*'

Secondly, by global fitting of the transient absorption spectra, we obtain the time constant of the dynamic process of the high-energy excited state before and after electrical stimulation, that is 'three decay components (340 fs, 20.6 ps and 5.82 ns for DtBuCzB) and two decay components (265 fs and 1.83 ns for complex after electrical stimulation) can be obtained.' By assigning the time constant, the relaxation process of high-energy excited states can be obtained, and the luminous mechanism of materials can be understood. As mentioned in the manuscript '*There was no interaction between DtBuCzB and p-BQ in the initial state, and the optical properties of the system were mainly contributed by DtBuCzB. Molecule transitioned from the ground state to the excited state, went through internal conversion, intersystem crossing, and finally returned to the ground state by radiative transition, along with fluorescence emission at 488 nm. When the electrical stimulation was applied, the electrical responsive B–O coordination bond was formed, and the energy level of complex has been changed. The excited-state electron went through vibrational relaxation, and then returned to the ground state by radiation transition, along with the blue shifted fluorescence emission at 382 nm.*' The above information is helpful for people to better understand the luminescence mechanism of B, N-PAHs, master the method of regulating their photophysical properties, and provide reference for the design of B, N-PAHs.

And the relevant description of the role of transient absorption spectrum

'The above information is helpful for people to better understand the luminescence mechanism of B, N-PAHs, master the method of regulating their photophysical properties, and provide reference for the design of B, N-PAHs.'

has been added in **Page 11 in Manuscript.**

Major comment 4:

The authors should consider mass spectrometry to elucidate the formation of the suggested assemblies. DOSY NMR spectroscopy could also be an alternative.

Response:

Thank you very much for this excellent suggestion to help our work more convincing. We have done the relevant experiments to verify the above issues in the revised manuscripts.

First, DtBuCzB were mixed with different proportions of hydroquinone di-anion (from hydroquinone di-lithium salt) for mass spectrometry. When a small amount of hydroquinone di-anion was added, $[\text{Li}(\text{DtBuCzB}\cdot p\text{-BQ})]^-$, $[\text{H}(\text{DtBuCzB}\cdot p\text{-BQ})]^-$ and $[(\text{DtBuCzB})_2\cdot p\text{-BQ}]^{2-}$ can be found in the mixed system, in which the di-coordination was dominant. When excessive hydroquinone di-anion was added, the presence of $[\text{Li}(\text{DtBuCzB}\cdot p\text{-BQ})]^-$ and $[(\text{DtBuCzB})_2\cdot p\text{-BQ}]^{2-}$ can still be observed in the mixed system, which is dominated by mono-coordination. The above phenomenon was consistent with the $^1\text{H-NMR}$ of DtBuCzB with different equivalents hydroquinone di-anion (*Comment 1*), which proved the formation of B–O coordination bonds in complex.

And, the new **Supplementary Fig. 11**

Supplementary Fig. 11. Mass spectrometry of DtBuCzB, and the mixtures of DtBuCzB and different equivalent hydroquinone di-lithium in THF recorded at room temperature.

has been added on **Page 20** of **Supplementary Information**.

And the Corresponding discussion

‘At the same time, DtBuCzB were mixed with different proportions of hydroquinone di-anion (from hydroquinone di-lithium salt) for mass spectrometry. As shown in **Supplementary Fig. 11**, when a small amount of hydroquinone di-anion was added, $[Li(DtBuCzB \cdot p-BQ)]^-$, $[H(DtBuCzB \cdot p-BQ)]^-$ and $[(DtBuCzB)_2 \cdot p-BQ]^{2-}$ can be found in the mixed system, in which the di-coordination was dominant. When excessive hydroquinone di-anion was added, the presence of $[Li(DtBuCzB \cdot p-BQ)]^-$ and $[(DtBuCzB)_2 \cdot p-BQ]^{2-}$ can still

be observed in the mixed system, which is dominated by mono-coordination. The above phenomenon was consistent with the $^1\text{H-NMR}$ of DtBuCzB with different equivalents hydroquinone di-anion, which proved the formation of B–O coordination bonds in complex.’

has been added on **Page 7** in **Manuscript**.

Meanwhile, **Supplementary Fig. 17** showed the representative $^1\text{H-DOSY}$ NMR spectra of the mixture. It can be found that the diffusion coefficient of hydroquinone di-anion in the mixed system with DtBuCzB was lower than the corresponding value of the original free hydroquinone di-anion. These changes were directly related to the formation of dynamic interactions in this system (*Carbohydrate Polymers*, 2018, 198, 294–301; *Eur. J. Inorg. Chem.*, 2022, 202100842).

And, the new **Supplementary Fig. 17**

Supplementary Fig. 17. Diffusion ordered spectroscopy (DOSY, 600 MHz) of free hydroquinone di-lithium, and the mixtures of DtBuCzB and 0.5 eq. hydroquinone di-lithium in *THF- d_8* recorded at room temperature.

has been added on **Page 26** of **Supplementary Information**.

And the corresponding discussion

‘The representative ^1H -DOSY NMR spectra of the mixture were shown in **Supplementary**

Fig. 17. It can be found that the diffusion coefficient of hydroquinone di-anion in the mixed system with DtBuCzB was lower than the corresponding value of the original free hydroquinone di-anion. These changes were directly related to the formation of dynamic interactions in this system.^{S8-S9}

has been added in **Supplementary Information**.

And the relevant Reference

‘8. Kasprzak, A., Borys, K. M., Molchanov, S. & Adamczyk-Woźniak, A. Spectroscopic insight into supramolecular assemblies of boric acid derivatives and β -cyclodextrin. *Carbohydr. Polym.* **198**, 294-301 (2018).

9. Rudlof, J. et al. Synthesis of bifunctional boron - lewis acids – thorough investigation of the adduct formation with pyrimidine. *Eur. J. Inorg. Chem.* **2022**, 202100842 (2022).’

has been added in the revised **Supplementary Information**.

Major comment 5:

The authors mentioned that “Compared with the above stimulation methods, there are few researches on dynamic covalent bonds in response to electrical stimulation, which have advantages of good controllability.” There are a number of reports on redox-switchable coordination chemistry (J. Am. Chem. Soc. 2018, 140, 14590–14594). Additionally, I also wonder the authors have considered electrochemically switchable molecular machines.

Response:

We sincerely thank you for your valuable input, which helped us to realize in time that some important references to our work in related areas were missing from our original manuscript. In the revised version, we have reviewed the literature and supplemented the relevant previous work. The details are as follows.

So, the revised discussion in **Introduction** on **Page 2**

‘Compared with the above stimulation methods, electricity-responsive bonds have the advantages of good controllability, and have been well studied into catalytic reactions based on metal coordination complexes,^[50, 51] artificial molecular machines,^[53-55] molecular switches,^[56] electrochromism^[57] and electrofluorochromism^[58,59] and other fields. However, there are few researches on electrical regulation of B bonds.’

has replaced the original

‘Compared with the above stimulation methods, there are few researches on dynamic covalent bonds in response to electrical stimulation, which have advantages of good controllability.’

And, the corresponding references

‘51. Cheng, H. F., d’Aquino, A. I., Barroso-Flores, J. & Mirkin, C. A. A redox-switchable, allosteric coordination complex. *J. Am. Chem. Soc.* **140**, 14590-14594 (2018).

52. Mitchell, B. S. *et al.* Redox-switchable allosteric effects in molecular clusters. *JACS Au* **2**, 92-96 (2021).

53. Nijhuis, C. A., Ravoo, B. J., Huskens, J. & Reinhoudt, D. N. Electrochemically controlled supramolecular systems. *Coordin. Chem. Rev.* **251**, 1761-1780 (2007).

54. Schröder, H. V. & Schalley, C. A. Tetrathiafulvalene – a redox-switchable building block to control motion in mechanically interlocked molecules. *Beilstein J. Org. Chem.* **14**, 2163-2185 (2018).

55. Yanilkin, V. V. & Stepanov, A. S. Electrochemistry of macrocyclic compounds: redox-switchable molecular systems. *J. Iran. Chem. Soc.* **20**, 257-289 (2022).

56. Harvey, E. C., Feringa, B. L., Vos, J. G., Browne, W. R. & Pryce, M. T. Transition metal functionalized photo- and redox-switchable diarylethene based molecular switches. *Coordin. Chem. Rev.* **282-283**, 77-86 (2015).

57. Wang, Y. *et al.* A see-through electrochromic display via dynamic metal-ligand interactions. *Chem* **7**, 1308-1320 (2021).

58. Kanazawa, K., Nakamura, K. & Kobayashi, N. High-contrast electroswitching of emission and coloration based on single-molecular fluoran derivatives. *J. Phys.Chem. A* **118**,

6026-6033 (2014).

59. Kanazawa, K., Nakamura, K. & Kobayashi, N. Electroswitchable optical device enabling both luminescence and coloration control consisted of fluoran dye and 1,4-benzoquinone. *Sol. Energy Mater. Sol. Cells* **145**, 42-53 (2016).’

have been added in **References**.

Major comment 6:

I suggest “electric switchable B-O bond” to be revised as “electrochemically switchable B–O bond”

Response:

Thank you for your very helpful and professional advice on improving our work, so that our article can be expressed more accurately. For more accurate expression, we changed the ‘electric switchable B-O bond’ to ‘electrochemically switchable B–O bond’, and modified the corresponding parts of the manuscript.

For example,

Title ‘An electrical responsive B-O dynamic bond to switch photoluminescence of boron-nitrogen-doped polyaromatics’ has been changed to

‘An electrochemically responsive B–O dynamic bond to switch photoluminescence of boron-nitrogen-doped polyaromatics’;

And, ‘An electrical responsive B-O dynamic coordination bond was proposed, and used to regulated the photophysical processes of boron-nitrogen-doped polyaromatic hydrocarbons (B,N-PAHs).’ in **Abstract** has been changed to

‘An electrochemically responsive B–O dynamic coordination bond was proposed, and used to regulated the photophysical processes of boron-nitrogen-doped polyaromatic hydrocarbons (B,N-PAHs).’;

And, ‘Herein, an electrical responsive reversible B-O dynamic coordination bond was

proposed' in **Introduction** has been changed to

'Herein, an electrochemically responsive reversible B–O dynamic coordination bond was proposed';

And, 'In summary, we demonstrated a new electrical responsive B-O dynamic coordination bond,' in **Conclusion** has been changed to

'In summary, we demonstrated a new electrochemically responsive B–O dynamic coordination bond';

And, the rest 'electric switchable B-O bond' in the manuscript has been modified to 'electrochemically switchable B–O bond'.

Major comment 7:

I am curious about the extent to which this strategy is applicable beyond this specific example (i.e., scope). If a similar approach is feasible for other boron-doped PAHs, including the well-known DABNA, it would attract broader interest.

Response:

We deeply appreciate your valuable suggestion to help us expand the audience area of our work. In the revised manuscript, the classical DABNA was used to explore the universality of electrochemically regulated coordination bonds used to regulate the photophysical properties of B, N-PAHs. As shown in **Supplementary Fig. 15**, the photophysical properties of the mixed system consisting of DABNA and *p*-BQ can be reversibly regulated under electrical stimulation.

Accordingly, a new **Supplementary Fig. 15** was added on **Page 24** of **Supplementary Information**.

Supplementary Fig. 15 The emission of the mixture of *p*-BQ ($1.0 \times 10^{-3} \text{ mol L}^{-1}$) and DABNA ($1.0 \times 10^{-4} \text{ mol L}^{-1}$) in THF with $1.0 \times 10^{-1} \text{ mol L}^{-1}$ tetrabutylammonium hexafluorophosphate (TBAPF₆) before and after electrical stimulation.

And the corresponding description

‘The universality of electrochemically regulated B–O coordination bonds had been explored further using the well-known DABNA (**Supplementary Fig. 15**). The photophysical properties of the mixed system consisting of DABNA and *p*-BQ can be reversibly regulated under electrical stimulation.’

has added on **Page 12** to the **Manuscript**.

Minor comment 1:

Rather than hyphen (-), please use en dash(–) to express minus sign or chemical bond.

Response:

Thank you for this suggestion to make our expression more accurate. We have replaced the corresponding hyphen (-) in the article with en dash (–) to express minus sign or chemical bond.

For example,

‘-1.6 V’ has been modified as ‘–1.6 V’

And ‘B-O bond’ has been modified as ‘B–O bond’, etc.

Minor comment 2:

In “p-BQ”, “p” should be in italic.

Response:

Thank you very much for carefully correcting our mistakes. we have corrected it.

Minor comment 3:

In Fig. 2, please explain what E1–E6 means in detail.

Response:

Thanks for your helpful advice to make our manuscript easier to understand. Herein, the relevant explanation ‘With the addition of electron acceptor (DtBuCzB), the interaction occurred between them, and various possible reaction processes are depicted in **Fig. 3b**, where E1 to E6 correspond to the voltage value of the reduction potential of each reaction process respectively.’ has been added in **Page 8 of Manuscript**.

Minor comment 4:

In the caption for Table 1, the association constant “Ka” should be revised with “a” as subscript.

Response:

Thanks a lot for your careful checks. We apologize for our carelessness. Based on your comments, we have revised it.

And the revised **Table 1** on **Page 6** in **Manuscript**

Table 1. Association constant K_a value^a of DtBuCzB and *p*-BQ system before and after electrical stimulation at 298 K.

	DtBuCzB and p -BQ	DtBuCzB and p -BQ under electricity
K_a/M^{-1}	210 ± 7.1	26000 ± 5700

^aThe K_a value is obtained from the emission spectrum in THF.

has been used to replace the original one.

Minor comment 5:

Page 7, Paragraph 1, Line 3: Table S1 -> Supplementary Table 1.

Response:

We apologize for our careless mistakes. Thank you for your reminder. Herein, ‘Compared with ... listed in **Table S1**’ has been changed to ‘As shown in **Supplementary Table 1**,’ in **Page 9** of **Manuscript**.

Minor comment 6:

Page 8, Paragraph 2, Line 8: radiation transition -> radiative transition.

Response:

Thank you for carefully correcting our mistakes.

Herein, ‘and then returned to the ground state by radiation transition’ has been changed to ‘and then returned to the ground state by radiative transition’ in **Page 11** of **Manuscript**.

Minor comment 7:

In References (including Supplementary References), the journal name for “Angew. Chem. Int. Ed.” should be revised as “Angew. Chem. Int. Ed.”.

Response:

Thanks a lot for your very careful reading. We apologize for our carelessness. In our resubmitted manuscript, the corresponding content was revised. Herein, **Reference 18, 26, 30, 36, 40, 43, 65, 69, 73, and Supplementary Reference 5** in the revised version have been corrected into ‘*Angew. Chem. Int. Ed.*’.

Minor comment 8:

In Supplementary Fig. 4, “Modle” should be revised to “Model”, and “NMR Titradition” should be revised to “Fluorescence Titration”. Please revise the caption for Supplementary Fig. 4b as well. It is confusing as (a) and (b) is essentially the same.

Response:

Thanks for your careful checks. We apologize for our carelessness. Herein, the revised **Supplementary Fig. 6. on Page 15 of Supplementary Information**

Supplementary Fig. 6. The non-linear curve-fitting (fluorescence titrations) for the complexation of (a) DtBuCzB ($1.0 \times 10^{-5} \text{ mol L}^{-1}$) and *p*-BQ without electrical stimulation, and (b) DtBuCzB ($1.0 \times 10^{-5} \text{ mol L}^{-1}$) and *p*-BQ under electrical stimulation in THF at 298 K.

has replaced the original

Supplementary Fig. 4. The non-linear curve-fitting (emission spectra titrations) for the complexation of DtBuCzB ($1.0 \times 10^{-5} \text{ mol L}^{-1}$) and (a) *p*-BQ and (b) *p*-BQ⁻ in THF at 298 K.

Minor comment 9:

The caption for Supplementary Fig. 14 does not contain any information about temperature, phase (solution or solid), waveband (excitation energy), and etc.

Response:

Thanks a lot for your comprehensive and detailed suggestions. Herein, the caption of

Supplementary Fig. 24 has been attached with experimental details.

The revised **Supplementary Fig. 24** on **Page 33**

Supplementary Fig. 24. EPR of PTMA-co-BP recorded under the condition of FrequencyMon was 9.417976 GHz, BridgeCalib was 61.8, Power was 0.06325 mW, and PowerAtten was 35.0 dB in solid phase at room temperature.

has replaced the original one.

To reviewer #3:

Comment:

Response:

Dear reviewer, thank you very much for taking your valuable time to review our manuscript very carefully. We sincerely thank you for your valuable feedback that we have used to improve the quality of our manuscript. And, we will try our best to revise our manuscript according to your suggestions. We also look forward to more opportunities to receive your work guidance and possible cross-collaboration in future.

To reviewer #4:

Comment:

The manuscript by Yang et al. reports on the modulation of the photophysical properties of a boron-nitrogen-doped polyaromatic hydrocarbons (DtBuCzB) through a combination of complexation chemistry and electrochemical processes. In particular, complexation of DtBuCzB with p-Benzoquinone (p-BQ) via the formation of a B-O bond, is modulated by the electrochemical reduction of p-BQ. The result is a system (it can be considered a dyad) that switches from one absorption/fluorescence state to another upon electrochemical reduction. Very interesting is the fluorescence switch from green (emission at 488 nm) to blue (emission at 382 nm) with more than 100 nm color modulation. This electrofluorochromic effect has been well supported by a mechanistic point of view, corroborated by both experiments and theory. I believe that the manuscript is novel and it can pave the way for the design of new materials based on dynamic switching bond. It is worth to be published after some revisions according to the comments appended below.

Response:

Dear reviewer, thank you very much for taking your valuable time to review our manuscript very carefully. From all your questions and suggestions, we realize that we are very lucky to have such a deep-thinking and meticulous top scientist in the field to help us improve the quality of our articles. Thank you very much for your professional comments and very valuable suggestions on our article. As you are concerned, there are several problems that need to be addressed. According to your suggestions, we have made extensive corrections to our previous draft, the detailed corrections are listed below.

Major comment 1:

The authors should show the effect of the electric stimulation on DtBuCzB at voltages < -1.66 V at which reduction of the DtBuCzB occurs.

Response:

We deeply appreciate your valuable suggestion to help us our work be more comprehensive. Based on your professional advice, the related experiments have been used to investigate changes in photophysical properties of individual molecules during electrochemical reduction. As shown in **Supplementary Fig. 5**, the photophysical properties of individual DtBuCzB under negative voltage stimulation also changed significantly, due to the change in the density of the electron density around B atom. However, under the stimulation of reverse voltage, its photophysical properties cannot be reversibly returned to the initial state. At the same time, the fluorescence changes at 488 nm and absorbance changes at 467 nm of individual DtBuCzB were agreement with this. We speculated that this might be caused by irreversible side reactions. In contrast, the mixed system consists of DtBuCzB and *p*-BQ has the advantage of low redox potential and good reversibility, which is due to the excellent electrochemical redox property of *p*-BQ.

Supplementary Fig. 5. The spectra of (a, c) emission and (b, d) absorption spectra of alone DtBuCzB (1.0×10^{-4} mol L $^{-1}$) in THF with 1.0×10^{-1} mol L $^{-1}$ TBAPF $_6$ when the solutions were added 0 V, -1.5 V/ -1.8 V and $+0.2$ V in-situ, ex = 350 nm. Changes in (e) emission at 488 nm (top) and (f) absorption spectra at 467 nm (top), during cyclic voltammograms (CVs, bottom) in situ of DtBuCzB (1.0×10^{-4} mol L $^{-1}$) in THF with 1.0×10^{-1} mol L $^{-1}$ TBAPF $_6$, ex = 440 nm.

The new **Supplementary Fig. 5** has been added on **Page 14** to **Supplementary Information**, and related discussion ‘Compared with the irreversible electrochemical reduction of individual DtBuCzB (**Supplementary Fig. 5**), the hybrid system has the advantages of low redox potential and good reversibility.’ has been added on **Page 5** of **Manuscript**.

Major comment 2:

The coordination effect of DMAP on the optical properties of DtBuCzB has been shown as a function of the concentration of DMAP. Is there any similar concentration-dependent effect with *p*-BQ?

Response:

Thank you for taking valuable time to reflect on our work. Based on your excellent suggestion, we have added more related experiments. The experimental results showed that the concentration of *p*-BQ can affect the coordination effect between DtBuCzB and *p*-BQ. The concentration of benzoquinone anions increased when a fixed voltage was applied to the mixed system, along with more obvious changes in photophysical properties in the system. Therefore, it can be concluded that the coordination effect was related to concentration.

Supplementary Fig. 16. The emission (a) and absorption spectra (b) of the mixture of *p*-BQ ($1.0 \times 10^{-3} \text{ mol L}^{-1}$) and DtBuCzB ($1.0 \times 10^{-4} \text{ mol L}^{-1}$) in THF with $1.0 \times 10^{-1} \text{ mol L}^{-1}$ TBAPF₆ when the solutions were added -0.9 V in situ with different time.

The new **Supplementary Fig. 16** and related discussion ‘The experimental results showed that the concentration of *p*-BQ can affect the coordination effect between DtBuCzB and *p*-BQ. As shown in **Supplementary Fig. 16**, the concentration of benzoquinone anions increased when a fixed voltage was applied to the mixed system, along with more obvious changes in photophysical properties in the system. Therefore, it can be concluded that the coordination

effect was related to concentration.’ have been added on **Page 25** of **Supplementary Information**.

Major comment 3:

In Supplementary Fig. 7 the effect of phenolate on the optical properties of DtBuCzB has been shown. However, the authors should show the entire emission spectral region as in Fig. 1b in order to understand whether the peak at 375 nm still arise upon quenching the one at 488 nm, as in the case of p-BQ.

Response:

We sincerely appreciate the valuable comments. As you suggested, an extended range emission spectrum has been tested to investigate the spectral changes after the addition of the reference sodium phenol. The experimental results showed that the change trend of photophysical properties after the addition of phenol anion was consistent with the electrochemical reduction of p-BQ. Specifically, the initial absorption and emission characteristic peaks of DtBuCzB were quenching, and both of them are blue shifted, which was caused by the conversion of B from sp² to sp³ hybridization.

The revised **Supplementary Fig. 3** on **Page 12**

Supplementary Fig. 3. (a) Emission and (b) absorption spectra of DtBuCzB (1.0×10^{-5} mol L⁻¹) in THF with adding sodium phenolate (1.5×10^{-5} mol L⁻¹), ex = 350 nm.

has replaced the original

Supplementary Fig. 7. (a) Absorption and (b) emission spectra of DtBuCzB ($1.0 \times 10^{-5} \text{ mol L}^{-1}$) in THF with adding different concentration of sodium phenolate, $\text{ex} = 440 \text{ nm}$.

Major comment 4:

The electrochemical mechanism should be better discussed. Specifically, the mechanistic explanation provided to justify the disappearance of the second reduction peak of pBQ, the concomitant increase of the intensity of the first reduction peak and the occurrence of peak IIIa, is not fully convincing. The authors stated that “It indicated that the complex of p-BQ•- and DtBuCzB was relatively easier to be further reduced than p-BQ alone, due to the electron-absorbing effect of the DtBuCzB.” However, I believe that a change in the redox potential rather than a change of the intensity of the Ic redox peak should be observed. Indeed, the half-wave oxidation peak of the $\text{pBQ/DtBuCzB}]^{2-}$ complex is at higher potential than that corresponding to the pBQ alone. However, the corresponding half-wave reduction peak is not observed in the CV. Therefore, it seems that complexation occurs only after pBQ has been reduced to the monocation radical. Under these conditions, the second reduction process becomes progressively hampered due to complexation. When the complex is oxidized at higher potentials (half-wave peak IIIa), it is no longer stable and dissociates, releasing the free DtBuCzB which accumulates at the working electrode at each cycle, with the consequent increase of the intensity of the Ic peak.

Response:

Thanks for your very constructive suggestions. We are very happy to have the opportunity to discuss this section with you. The change of electrochemical redox voltage is very strong evidence for analyzing the interaction between DtBuCzB and *p*-BQ. After carefully understanding and analyzing your suggestion, we believe that the experiment of continuous scanning of the mixed system over and over again can provide a reference for the analysis of electrochemical processes. As shown in the figure below, when the mixed system consisted of *p*-BQ and 4 eq. DtBuCzB was continuously scanned for 20 cycles, the current intensity did not change significantly. This may indicate that the effect of BQ accumulation due to complex dissociation in each cycle is not large.

Fig. Multiturn cyclic voltammograms of the mixture consists of *p*-BQ (1.0×10^{-3} mol L $^{-1}$) with 4 eq DtBuCzB in THF with 1.0×10^{-1} mol L $^{-1}$ TBAPF $_6$.

Therefore, we tend to think that the main reason for the increase in the peak current of **Ic** is that, with the addition of DtBuCzB, the increase of the intensity of reduction peak **Ic** is due to the formation of complex generated by Process ① and ③, which is more easily reduced than the semiquinone radical, which is affected by the introduction of electron-absorbing effect of DtBuCzB that caused $E_4 > E_2$. Therefore, with the addition of DtBuCzB, Process ④ gradually replaces Process ②, making the potential of reduction peak **Iic** gradually less negative. And when a large amount of DtBuCzB was added, the **Iic** will have a fusion with the **Ic**, which may be the reason of the increase in the peak value of **Ic**. At the same time, the

reduction peak area proved this phenomenon (**Supplementary Fig. 12**). When the above phenomenon occurred, the area of reduction peak **Ic** is twice that of the original one observed in the absence of DtBuCzB. This behavior was also observed for the reduction of 5H-benzo[b]carbazole-6,11-diones (BCDs) in the presence of phenol as proton donors. (*Current Organic Chemistry*, 2004, 8, 1721-1738).

Supplementary Fig. 12. Cyclic voltammograms and peak area of (a) mixture of *p*-BQ (1.0×10^{-3} mol L^{-1}) with 4 eq DtBuCzB and (b) individual *p*-BQ (1.0×10^{-3} mol L^{-1}) in THF with 1.0×10^{-1} mol L^{-1} TBAPF₆. All the scan rates are 50 mV/s.

So, the new **Supplementary Fig. 12** has been added on **Page 21** to **Supplementary Information**. And, the related discussion on **Page 9** in **Manuscript** ‘With DtBuCzB added, the first reversible redox wave **Ic–Ia** gradually became irreversible, while the new irreversible half-wave **IIIa** emerged at more positive potential, accomplished by the second reversible reduction wave **Ic–IIa** became a less negative, wide wave and gradually disappeared. The above phenomenon was consistent with the electrochemical behavior of *p*-BQ derivative in the presence of proton donor acid⁷⁵⁻⁷⁷, which have proved that those phenomena was due to the complexation of *p*-BQ^{•−}/*p*-BQ^{2−} with H⁺ through the Process ①, ③, ④ and ⑦ (**Fig. 3b**). Herein, the emergency of new irreversible half-wave **IIIa** illustrated that complexation occurred only after *p*-BQ has been reduced to the semiquinone radical (*p*-BQ^{•−}) (Process ①, ③). Under these conditions, [DtBuCzB·*p*-BQ]^{•−} was more easily reduced than *p*-BQ^{•−}, which is affected by the introduction of electron-absorbing effect of DtBuCzB that caused $E_4 > E_2$.

Therefore, with the addition of DtBuCzB, Process ④ and ⑦ gradually replaced Process ②, making the potential of reduction peak **IIc** gradually less negative. And when a large amount of DtBuCzB was added, the **IIc** will have a fusion with the **Ic**, which may be the reason of the increase in the peak value of **Ic**. At the same time, the reduction peak area proved this phenomenon (**Supplementary Fig. 12**). This behavior was also observed for the reduction of 5H-benzo[b]carbazole-6,11-diones (BCDs) in the presence of phenol as proton donor.⁷⁴ The above phenomenon supported that the $[(\text{DtBuCzB})_2 \cdot p\text{-BQ}]^{2-}$ and $[\text{DtBuCzB} \cdot p\text{-BQ}]^{2-}$ complex is more favorable than $[\text{DtBuCzB} \cdot p\text{-BQ}]^-$ complex.'

has been used to replace the original 'With DtBuCzB added, the intensity of reduction peak **Ic** of *p*-BQ increased accordingly, The above phenomenon supported that the $[(\text{DtBuCzB})_2 \cdot p\text{-BQ}]^{2-}$ and $[\text{DtBuCzB} \cdot p\text{-BQ}]^{2-}$ complex is more favorable than $[\text{DtBuCzB} \cdot p\text{-BQ}]^-$ complex.'

Major comment 5:

The peculiarity of this system is that its fluorescence switches between two different colors (from green to blue) upon the application of the electrical stimulation. It is not simply an electrofluorochromic quenching process. Therefore, the full characterization of the device should include the response of the system even at 382 nm.

Response:

Thank you very much for your professional suggestions to help our work more systematic. Therefore, the fluorescence change of the EFC device was investigated in the expanded wavelength range, and it only showed the fluorescence quenching property at 490 nm, which was because the indium tin oxide (ITO) glass electrode used in the preparation of the device can block the propagation of the light signal within the range of less than 400 nm. Therefore, the dual-band switched EFC property of the material cannot be well displayed in the device, and the subsequent search for a suitable electrode may be a way. However, in solution, the EFC properties of two-band switching were investigated (**Fig. 1b and 1c**), its basic properties

were characterized (**Supplementary Table 3**), and the mechanism of its photophysical properties was explored (**Fig. 4c**).

Fig. Emission spectra of the device under different voltages, ex = 350 nm.

Major comment 6:

A full characterization of the photophysical properties of free DtBuCzB and its complexes at different oxidation states should be provided (quantum yields in solution and in device, etc.).

Response:

We sincerely appreciate the detailed and comprehensive suggestions to help our work more comprehensive. Based on your professional advice, the characterization of the photophysical properties of free DtBuCzB and its complexes at different oxidation states were provided. Since the above tests need to be carried out under completely closed conditions, we are sorry that it is difficult to directly test the EFC system under in-situ electrical stimulation. The relevant instruments and devices and experimental techniques will be also the targets of our subsequent efforts. Therefore, phenol anionic salts (sodium phenol and hydroquinone di-lithium salts, respectively) were selected to mimic the products after the electrochemical reduction of *p*-BQ. They are mixed with DtBuCzB to prepare the complexes of the system after electrical stimulation. The fluorescence quantum efficiency and lifetime were characterized under these conditions.

Among them, hydroquinone di-lithium salt was prepared by capturing the proton of hydroquinone by n-butyllithium. The reason for choosing n-butyllithium was that it has a larger solubility difference with the product (salt), so that product was easy to purify. The detailed procedure is as follows: a stock solution of 2 eq. n-butyllithium (2.5 mol/L in n-hexane) was added to a solution of hydroquinone (0.1 mol/L in THF) drop by drop under a nitrogen atmosphere at -78°C . After the reaction was completed, the solid was filtered, and the solid was repeatedly washed with anhydrous THF to remove the unreacted n-butyllithium, and the final product was obtained. $^1\text{H-NMR}$ and FT-IR spectra are shown in the **Supplementary Fig. 19 and Fig. 22**.

Supplementary Fig. 19. $^1\text{H-NMR}$ spectra of hydroquinone and the product (hydroquinone dilithium) from hydroquinone and n-butyllithium in THF- d_8 recorded at 400 MHz at room temperature.

Supplementary Fig. 22. FT-IR spectra of hydroquinone (H₂Q) and hydroquinone dilithium.

Compared with the initial hydroquinone, the characteristic peak of the product disappeared at 7.5 ppm, indicating that the proton hydrogen of hydroquinone was robbed by the strong organic base (n-butyllithium). In FT-IR spectra, hydroquinone showed the characteristic peak at 3261 cm⁻¹ and 1353 cm⁻¹, corresponding to the stretching vibration and bending vibration of -OH, respectively. Those peaks disappeared in hydroquinone di-lithium salt, which proved the deprotonation was complete.

Based on the above fact, the phenol anionic salts (sodium phenol and hydroquinone di-lithium salts, respectively) were mixed with DtBuCzB to prepare the complexes of the system after electrical stimulation to characterize the photophysical properties complexes.

Therefore, the **Supplementary Table 3** in **Page 36** has been added in the revised **Supplementary Information**.

Supplementary Table 3. The summary of physical properties of DtBuCzB (sp²-hybrid), DtBuCzB·OC₆H₅⁻ (sp³-hybrid), and DtBuCzB·O₂C₆H₄²⁻ (sp³-hybrid) in solution.

Solution	$\Phi_{\text{PL, ex=440 nm}}$	$\Phi_{\text{PL, ex=340 nm}}$	τ_p [ns]
DtBuCzB	91.49% (488 nm)	-	5.25
DtBuCzB·OC₆H₅⁻	0.35% (488 nm)	21.12% (368 nm)	3.85
DtBuCzB·O₂C₆H₄²⁻	0.18% (488 nm)	35.26% (366 nm)	3.70

And, the corresponding discussion

‘The basic properties of fluorescence, fluorescence quantum efficiency (Φ_{PL}) and fluorescence lifetime (τ), of the system in different states were characterized with sodium phenol (NaOC₆H₅) and hydroquinone di-lithium salt (Li₂O₂C₆H₄) as mimics of product generated by the electro-reduction of *p*-BQ, and the results were listed in **Supplementary Table 2**. With the addition of OC₆H₅⁻ or O₂C₆H₄²⁻, the molecular structure will change due to the coordination effect. And the Φ_{PL} of DtBuCzB at 488 nm dropped sharply to near zero, accompanied by the appearance of a new fluorescence peak near 366 nm or 368 nm.’

has been added on **Page 11** in the revised **Manuscript**.

And for EFC device, the above information was summarized into **Supplementary Table 2** and added into **Page 35** in the revised **Supplementary Information**.

Supplementary Table 2. The summary of basic performance of EFC device.

	λ_{em} [nm]	$\Phi_{\text{PL}}^{\text{a)}$	$\Phi_{\text{PL}}^{\text{b)}$	Contrast	η	Switching time [s]
						t_{on} : 0.4
Device	490	16.7%	1.0%	10.2	90%	t_{off} : 0.8

^{a)} Fluorescence quantum yield of initial device. ^{b)} Fluorescence quantum yield of device after electrical stimulation.

And, the corresponding discussion

‘And, the basic properties of EFC device before and after electrical stimulation were listed in **Supplementary Table 2**.’

has been added on **Page 13** in the revised **Manuscript**.

Major comment 7:

Finally, photographs of the device in the different states should be provided.

Response:

Thanks a lot for your very helpful suggestions. Photographs of EFC device in the different states were provided, which has been added in **Fig. 5d**.

Fig. 5 (d) The actual photos of the device in daylight and UV light.

Minor comment 1:

In the introduction, please, take into account a very important work in the field of molecular structure changes induced by electrical stimuli, that is Kanazawa et al J. Phys. Chem. A 2014, 118, 6026–6033 doi.org/10.1021/jp5060588 which investigate changes in the photophysical properties induced by electrical stimuli in a fluoran-based molecule.

Response:

We sincerely appreciate your valuable feedback, which made us realize in a timely manner that some reference materials related to relevant field work were omitted in the original manuscript. In the revised version, we have reviewed the literature and supplemented the relevant previous work. The details are as follows.

So, the revised discussion in **Introduction** on **Page 2**

‘Compared with the above stimulation methods, electricity-responsive bonds have the advantages of good controllability, and have been studied into catalytic reactions based on metal coordination complexes,^[50, 51] artificial molecular machines,^[53-55] molecular switches,^[56] electrochromism^[57] and electrofluochromism^[58,59] and other fields. However, there are few researches on electrical regulation of B bonds.’

has replaced the original

‘Compared with the above stimulation methods, there are few researches on dynamic covalent bonds in response to electrical stimulation, which have advantages of good controllability.’

Especially, the corresponding references

‘58. Kanazawa, K., Nakamura, K. & Kobayashi, N. High-contrast electroswitching of emission and coloration based on single-molecular fluoran derivatives. *J. Phys.Chem. A* **118**, 6026-6033 (2014).

59. Kanazawa, K., Nakamura, K. & Kobayashi, N. Electroswitchable optical device enabling both luminescence and coloration control consisted of fluoran dye and 1,4-benzoquinone. *Sol. Energy Mater. Sol. Cells* **145**, 42-53 (2016).’

have been added in **References**.

Minor comment 2:

The first part of Results and discussion up to ...electrochemical redox performance" should be moved in the introduction.

Response:

Thanks for your constructive suggestions to make our manuscript more logical. The relevant description 'B,N-PAHs have both desirable properties such as high color purity, ... electrochemical redox performance⁶²⁻⁶⁴.' has been moved into the Introduction.

Minor comment 3:

Fig. 4 d. the pulse sequence is -1.5V for 2s and +1.5V for 1.5 but it seems that between one and the other there is a pause of over 10 s OFF. What is the potential in these 10 s? 0V or is it held at +1.2V?

Response:

We really appreciate your careful advice. The pause is the 'Open-circuit voltage' which is used to balance the charge from electrical stimulation. And the relevant state was added to the caption of **Fig. 5f**. So, '(f) The stability of the device at 490 nm under 1000 cycles, stimulated with -1.5 V 2.0 s for quenching, open-circuit voltage, and +1.2 V 1.5 s for recovering (ex = 440 nm)' is the revised one.

Minor comment 4:

The authors report quantitative data on the EFC device such as number of cycles, ON and OFF times, % contrast lost, but do not report the maximum contrast value. This could be added.

Response:

Thanks for your helpful suggestions. We think this is an excellent suggestion. We have added the maximum contrast value as a performance indicator. In detail, ‘**And the fluorescence quenching efficiency was greater than 90%, and the maximum contrast value is 10.2.**’ has replaced the original ‘**And the fluorescence quenching efficiency was greater than 90%.**’ on **Page 13** in revised **Manuscript**.

To reviewer #5:

Comment:

I co-reviewed this manuscript with one of the reviewers who provided the listed reports as part of the Nature Communications initiative to facilitate training in peer review and appropriate recognition for co-reviewers.

Response:

Dear reviewer, thank you very much for taking your valuable time to review our manuscript very carefully. We sincerely thank you for your valuable feedback that we have used to improve the quality of our manuscript. And, we will try our best to revise our manuscript according to your suggestions. We also look forward to more opportunities to receive your work guidance and possible cross-collaboration in future.

REVIEWERS' COMMENTS

Reviewer #2 (Remarks to the Author):

I was pleased to review this revised manuscript, where the authors have adequately addressed my previous concerns. Particularly, with the revised Fig. 2 and Supplementary Fig. 11, I am convinced of the existence of a dynamic B–O bond. The additional experiment with DABNA highlights the universality of using B-O coordination bonds to achieve an electrofluorochromic system.

I recommend the publication of this manuscript in Nature Communications after addressing several minor corrections:

1. In Supplementary Fig. 17, the authors should replace triphenylborane with the structure of DtBuCZB, as it is confusing.
2. When referring to the hydroquinone dianion, the authors use both "di-anion" and "bi-anion." Please standardize the terminology and use one consistently.
3. Some figures that were added during the revision process, such as Supplementary Fig. 9 and 10, should be excluded from the manuscript. They do not contain crucial information but instead contribute to reader confusion. It would have been preferable if these figures were included as review-only figures.

Reviewer #3 (Remarks to the Author):

Reviewer #4 (Remarks to the Author):

The authors have greatly improved the manuscript following the criticisms of all the reviewers. There are just a few issues that should be addressed in order to better describe and present the overall work.

A general note about the terminology used by the authors. They always talk about electrical stimulation as to indicate that the system is responsive to an external electric stimulus. However, a better terminology should refer to the electrochemical reduction of p-BQ which leads to the EFC effect.

Another observation is on the value of the negative voltage used. Since the EFC effect is caused by the association of DtBuCzB with the reduced p-BQ, a value corresponding to the reduction potential of the latter would be sufficient to exploit the effect. Therefore, the authors should explain why they used the exact value of -0.9 V to investigate the EFC effect (see the next comment).

Regarding the electrochemical processes involved in the EFC effect (Fig. 3), it is clear that there is a superposition of the free and complexed species in solution, at least approximately up to a p-BQ:DtBuCzB equivalent ratio of 1:2.0 (dimensionless). It is also clear that complexation of the mono-anion radical causes a shift to lower values (due to the electrode deficiency of the boron compound) of the second reduction potential of p-BQ in the form of the monocoordinated species, which tends to be close to the first redox potential I_c , thereby increasing the intensity of the peak I_c . As a consequence, the corresponding half-wave oxidation peak (IIa) progressively decreases in intensity with the increase of the above ratio. On the other hand, the half-wave oxidation peak (Ia) corresponding to the peak I_c , is shifted to higher values with respect to that of the free p-BQ (Ia), leading to the peak IIIa. However, the peak I_c remains stable because it always corresponds to the first reduction process of the free p-BQ, since after the process IIIa the neutral benzoquinone is produced again and it is not significantly associated with DtBuCzB. The broadening of the redox processes may be due to the fact that slightly different complexed species may originate from the p-BQ and DtBuCzB interaction.

Therefore, it seems that the mono-coordinated species exhibits a reduction potential very close to that of the free p-BQ. Once the equivalents of DtBuCzB exceed a certain value (above ca. a 1:2.0 p-BQ:DtBuCzB equivalent ratio), the electrochemical behavior is determined by the free p-BQ in reduction (peak I_c) and the coordinated species in oxidation (peak IIIa), both broadened because they result from the superposition of the redox processes of the mono and di-coordinated species, which have slightly different redox potentials. Theoretically, a 1:2.0 p-BQ:DtBuCzB equivalent ratio should correspond to the saturation due to complete association between the di-anion and DtBuCzB.

A differential pulse voltammetry analysis of peak I_c and IIIa may help in elucidating this analysis. Moreover, the area of the two peaks should be almost the same but this can be verified once the CV is extended to cover the entire process IIIa.

In view of these explanation, the value of -0.9 V is justified because at this potential both the mono and dianion radicals are generated, leading also to a justification of the structural scheme presented in Fig. 4c, showing the interaction between the di-anion radical species with two molecules of DtBuCzB. This issue should be clearly highlighted.

Reviewer #5 (Remarks to the Author):

"I co-reviewed this manuscript with one of the reviewers who provided the listed reports. This is part of the Nature Communications initiative to facilitate training in peer review and to provide appropriate recognition for Early Career Researchers who co-review manuscripts."

Point-by-point Response to the Reviewers' Comments

Dear Editor and reviewers,

Here, the detailed point-by-point responses to the comments of reviewers' have been enclosed below.

To reviewer #2:

Comment:

I was pleased to review this revised manuscript, where the authors have adequately addressed my previous concerns. Particularly, with the revised Fig. 2 and Supplementary Fig. 11, I am convinced of the existence of a dynamic B–O bond. The additional experiment with DABNA highlights the universality of using B–O coordination bonds to achieve an electrofluorochromic system. I recommend the publication of this manuscript in Nature Communications after addressing several minor corrections:

Response:

Dear reviewer, thank you very much for your time and effort to carefully review the manuscript and provide valuable guidance. We are deeply gratified and motivated to receive the enthusiastic help and high recognition of the top experts in this field carefully selected by the editor. We also look forward to more opportunities to receive your work guidance and possible cross-collaboration in future. At the same time, we will strive to further improve the quality of this work according to other reviews input to benefit more readers. Thanks again for your help.

Comment 1:

In Supplementary Fig. 17, the authors should replace triphenylborane with the structure of DtBuCZB, as it is confusing.

Response:

Thank you for this suggestion to make our expression more accurate. We have replaced the triphenylborane with DtBuCzB in the original **Supplementary Fig. 17**. And the new **Supplementary Fig. 15** has been added on **Page 17** of **Supplementary Information**.

Supplementary Fig. 15. Diffusion ordered spectroscopy (DOSY, 600 MHz, THF-d_3) of free hydroquinone di-lithium, and the mixtures of DtBuCzB and 0.5 eq. hydroquinone di-lithium recorded at room temperature.

Comment 2:

*When referring to the hydroquinone dianion, the authors use both "di-anion" and "bi-anion."
Please standardize the terminology and use one consistently.*

Response:

Thanks a lot for your very careful reading. We apologize for our carelessness. In our resubmitted manuscript, we have consistently referred to the hydroquinone dianion as "**di-anion.**"

For example,

"... and excess **bi-anions** were characterized by $^{11}\text{B-NMR}$ " on **Page 6** in **Manuscript** has been changed to "... and excess **di-anions** were characterized by $^{11}\text{B-NMR}$ ".

"The peaks IIc and IIa respectively corresponded to the reduction of $\text{p-BQ}^{\cdot-}$ further to the **bi-anions** (p-BQ^{2-}) ..." on **Page 8** in **Manuscript** has been changed to "The peaks IIc and IIa respectively corresponded to the reduction of $\text{p-BQ}^{\cdot-}$ further to the **di-anions** (p-BQ^{2-}) ...".

Comment 3:

Some figures that were added during the revision process, such as Supplementary Fig. 9 and 10, should be excluded from the manuscript. They do not contain crucial information but instead contribute to reader confusion. It would have been preferable if these figures were included as review-only figures.

Response:

Thank you very much for your constructive suggestions to make our article more readable. According to your suggestions, we have removed the original **Supplementary Fig. 9** and **10** from the **revised Supplementary Information**.

And, the corresponding discussion "**The structure of DtBuCzB before and after coordination was further confirmed by COSY $^1\text{H-NMR}$ (Supplementary Fig. 9).**" and "At the same time,

variable temperature $^1\text{H-NMR}$ (**Supplementary Fig. 10**) of a mixture of DtBuCzB and 2 eq. hydroquinone di-anion has been explored, and a new characteristic hydrogen signal appeared at low temperature." have also been excluded in the **revised Manuscript**.

To reviewer #3:

Comment:

Response:

Dear reviewer, thank you very much for taking your valuable time to review our manuscript. We appreciate your friendly efforts to improve the quality of our paper to be a higher level. And, we will try our best to revise our manuscript according to your suggestions. We also look forward to more opportunities to receive your work guidance and possible cross-collaboration in future.

To reviewer #4:

Comment:

The authors have greatly improved the manuscript following the criticisms of all the reviewers. There are just a few issues that should be addressed in order to better describe and present the overall work.

Response:

Dear reviewer, thank you very much for taking your valuable time to review our manuscript very carefully. Thank you very much for your professional comments and valuable suggestions on our article. As you are concerned, there are several problems that need to be addressed. According to your suggestions, we have made relevant corrections to our previous manuscripts, the detailed corrections are listed below.

Comment 1:

A general note about the terminology used by the authors. They always talk about electrical stimulation as to indicate that the system is responsive to an external electric stimulus. However, a better terminology should refer to the electrochemical reduction of p-BQ which leads to the EFC effect.

Response:

Thank you for your very helpful and professional advice on improving our work, so that our article can be expressed more accurately. For more accurate expression, we changed the "electrical stimulation" to "electrochemical stimulation", and modified the corresponding parts of the manuscript.

For example, "Herein, an electrochemically responsive reversible B–O dynamic coordination bond ... regulated by electrical stimulation" in **Introduction** has been changed to "Herein, an electrochemically responsive reversible B–O dynamic coordination bond ... regulated by electrochemical stimulation";

And, "In summary, we ... under electrical stimulation" in **Conclusion** has been changed to "In summary, we ... under electrochemical stimulation";

And, the description in **Fig. 1f, 4c** and **5a** has also been changed from "electrical stimulation" to "electrochemical stimulation";

And, the rest "electrical stimulation" in the manuscript has been modified to "electrochemical stimulation".

Comment 2:

Another observation is on the value of the negative voltage used. Since the EFC effect is caused by the association of DtBuCzB with the reduced p-BQ, a value corresponding to the reduction potential of the latter would be sufficient to exploit the effect. Therefore, the authors should explain why they used the exact value of -0.9 V to investigate the EFC effect (see the next comment).

Response:

We appreciate your valuable advice to make our work more logical. The exact value of -0.9 V was chosen for two main reasons. On the one hand, as you mentioned below, -0.9 V can produce both the mono and dianion radicals at the same time, which can give us a comprehensive view of species that may be included in the mixed system after electrochemical stimulation (details are shown in the comments below). On the other hand, because of the above reason, -0.9 V can provide the most obvious photophysical property switching effect.

Comment 3:

Regarding the electrochemical processes involved in the EFC effect (Fig. 3), it is clear that there is a superposition of the free and complexed species in solution, at least approximately up to a p-BQ:DtBuCzB equivalent ratio of 1:2.0 (dimensionless). It is also clear that complexation of the mono-anion radical causes a shift to lower values (due to the electrode deficiency of the boron compound) of the second reduction potential of p-BQ in the form

of the monocoordinated species, which tends to be close to the first redox potential I_c , thereby increasing the intensity of the peak I_c . As a consequence, the corresponding half-wave oxidation peak (IIa) progressively decreases in intensity with the increase of the above ratio. On the other hand, the half-wave oxidation peak (Ia) corresponding to the peak I_c , is shifted to higher values with respect to that of the free $p-BQ$ (Ia), leading to the peak $IIIa$. However, the peak I_c remains stable because it always corresponds to the first reduction process of the free $p-BQ$, since after the process $IIIa$ the neutral benzoquinone is produced again and it is not significantly associated with $DtBuCzB$. The broadening of the redox processes may be due to the fact that slightly different complexed species may originate from the $p-BQ$ and $DtBuCzB$ interaction.

Therefore, it seems that the mono-coordinated species exhibits a reduction potential very close to that of the free $p-BQ$. Once the equivalents of $DtBuCzB$ exceed a certain value (above ca. a 1:2.0 $p-BQ$: $DtBuCzB$ equivalent ratio), the electrochemical behavior is determined by the free $p-BQ$ in reduction (peak I_c) and the coordinated species in oxidation (peak $IIIa$), both broadened because they result from the superposition of the redox processes of the mono and di-coordinated species, which have slightly different redox potentials. Theoretically, a 1:2.0 $p-BQ$: $DtBuCzB$ equivalent ratio should correspond to the saturation due to complete association between the di-anion and $DtBuCzB$.

A differential pulse voltammetry analysis of peak I_c and $IIIa$ may help in elucidating this analysis. Moreover, the area of the two peaks should be almost the same but this can be verified once the CV is extended to cover the entire process $IIIa$.

Response:

Great thanks for your professional hint and reminder on our work. We are very pleased to have an agreement with you in understanding of the electrochemical processes of the system, and we appreciate the method (differential pulse voltammetry analysis) your proposed to further verify that "when saturated $DtBuCzB$ was added, the electrochemical behavior is determined by the free $p-BQ$ in reduction (peak I_c) and the coordinated species in oxidation (peak $IIIa$), both broadened because they result from the superposition of the redox processes of the mono and di-coordinated species, which have slightly different redox potentials". Since the coordination reaction is a dynamic equilibrium process, we use excess $DtBuCzB$ (4 eq.) to make the coordination

occur as completely as possible. The experimental results of the differential pulse voltammetry (DPV) analysis with higher sensitivity for the mixed systems show that, there are two peaks in the range of peak **IIIc** when excess DtBuCzB is added to it. These two peaks can be attributed to the oxidation peak of mono- and di-coordinated species. However, the differential pulse voltammetry of the reduction peak is still one peak, possibly because the free *p*-BQ and the mono-coordinated species have very similar reduction potentials (**Supplementary Fig. 10**). Furthermore, the above conclusion is verified by the fact that the CV integral areas of the two peaks are basically the same (**Supplementary Fig. 10**). These phenomena further confirm the accuracy of our understanding of electrochemical processes.

Accordingly, the new **Supplementary Fig. 10** was used to replace the original as more convincing evidence.

CV of *p*-BQ and excess DtBuCzB

DPV of *p*-BQ and excess DtBuCzB

Supplementary Fig. 10. (a) Cyclic voltammogram of mixture of *p*-BQ ($1.0 \times 10^{-3} \text{ mol L}^{-1}$) with excess DtBuCzB in THF with $1.0 \times 10^{-1} \text{ mol L}^{-1}$ TBAPF₆, and the peak area of peak Ic and IIIa. The scan rates are 100 mV/s. Differential pulse voltammograms (DPV) for (b) reduction peak Ic and (c) oxidation peak IIIa of mixture of *p*-BQ ($1.0 \times 10^{-3} \text{ mol L}^{-1}$) with excess DtBuCzB in THF with $1.0 \times 10^{-1} \text{ mol L}^{-1}$ TBAPF₆. Pulse amplitude of 50 mV, a pulse width of 50 ms, and scan rate: 25 mV/s.

And, the relevant discussion "At the same time, differential pulse voltammograms of oxidation peak IIIa and reduction peak Ic and, and their peak area proved this phenomenon. (Supplementary Fig. 10)" has been added on Page 8 of Manuscript to replace the original "At the same time, the reduction peak area proved this phenomenon (Supplementary Fig. 10)"

Comment 4:

In view of these explanation, the value of -0.9 V is justified because at this potential both the mono and dianion radicals are generated, leading also to a justification of the structural scheme presented in Fig. 4c, showing the interaction between the di-anion radical species with two molecules of DtBuCzB. This issue should be clearly highlighted.

Response:

Thank you very much for providing this very careful and professional advice on improving our work. In order to express more clearly, we have highlighted the reasons "which can generate both mono- and di-coordination complexes" for choosing -0.9 V on Page 4 when it was first mentioned in Manuscript.

To reviewer #5:

Comment:

Response:

Dear reviewer, thank you very much for taking your valuable time to review our manuscript very carefully. We sincerely thank you for your valuable feedback that we have used to improve the quality of our manuscript. And, we will try our best to revise our manuscript according to your suggestions. We also look forward to more opportunities to receive your work guidance and possible cross-collaboration in future.

Reviewers' Comments

Reviewer #4 (Remarks to the Author):

All the criticisms have been addressed. I recommend the publication of the paper.

Reviewer #5 (Remarks to the Author):
